# Bayesian hierarchical models for multivariate mixed responses with repeated measures: A case study in arterial occlusive disease

Endris Assen Ebrahim[1]*, Mehmet Ali Cengiz[2]

**1** Department of Statistics, College of Natural and Computational Sciences, Debre Tabor University, South Gondar, Ethiopia, **2** Department of Mathematics and Statistics, College of Science, Imam Mohammad Ibn Saud Islamic University (IMSIU), Riyadh, Saudi Arabia

* end384@gmail.com

## Abstract

Modeling repeated measures of arterial occlusive diseases, such as peripheral artery disease (PAD), using data with mixed-type outcomes poses unique challenges due to complex dependency structures and diverse distributional assumptions. This study proposes a comprehensive Bayesian hierarchical modeling framework for the simultaneous analysis of binary and continuous outcomes observed repeatedly within individuals. We focus on the methodological comparison of three major Markov Chain Monte Carlo (MCMC) Bayesian computational methods—Metropolis-Hastings, Gibbs sampling, and Hamiltonian Monte Carlo (HMC) that are suitable for hierarchical models without random effects, as well as those with random intercepts and slopes, by utilizing arterial occlusive disease (AOD) data that includes repeated leg measurements on 16 patients with a total of 256 observations. We evaluate model performance across multiple criteria, including the widely applicable information criterion (WAIC), Leave-one-out information criteria (LOO-IC), K-fold cross-validation (K = 10), the Bayesian information criterion (DIC), and the BIC information complexity (ICOMP). Our results reveal that the full random effects model estimated via HMC performed better and achieved higher predictive accuracy across the considered information criteria for this small-sample, historical dataset used for modeling applications. This work emphasizes the importance of model selection strategies in hierarchical Bayesian analysis and highlights the advantages of employing modern MCMC techniques in medical applications. However, we realize that these findings may depend on the precise priors and parameterizations used and may not apply to all small-sample hierarchical datasets. Thus, expanding this model to larger, contemporary datasets will improve its generalizability and clinical relevance.

**Data availability statement:** The data were obtained from https://doi.org/10.2307/2532336. The extracted data used in this study are available at https://doi.org/10.5281/zenodo.17621089.

**Funding:** This work was supported and funded by the Deanship of Scientific Research at Imam Mohammad Ibn Saud Islamic University (IMSIU) (grant number IMSIU-DDRSP2601).

**Competing interests:** The authors have declared that no competing interests exist.

**Abbreviations:** AOD: Arterial Occlusion Diseases, BHM: Bayesian Hierarchical Modeling, BRMS: Bayesian regression models using 'Stan' (R package), CI: Credible Interval, DIC: Bayesian deviation criterion, EAS: Expected Value Posterior Estimation, ESS: Effective sample sizes, HMC: Hamiltonian Monte Carlo, HPD: 95% High Posterior Density(Credible)Interval, ICOMP: Bozdogan's Information Complexity/Criteria Index, LKJ: Lewandowski-Kurowicka-Joe, LOO-IC: Leave One-Out Information criterion, MCMC: Markov Chain Monte Carlo, MCMCglmm: Generalized Linear Mixed Models with MZMC (R package), R2MLwiN: A package for running MLwiN from within R using the Metropolis-Hastings algorithm, RCP: Reduced in cuff pressure measurement, WAIC: Watanabe-Akaike/Bayesian Information Criterion.

## 1. Introduction

Repeated measures with multivariate mixed outcomes—comprising both categorical and continuous response types—are prevalent in many fields, including medicine, social sciences, and engineering. Such data structures require sophisticated modeling strategies that account for intra-subject correlations and the complexities of joint distributions. Traditional univariate approaches often fail to exploit correlations across outcomes, leading to inefficient parameter estimation [1]. Hierarchical Bayesian models provide a flexible framework for addressing these challenges by incorporating prior knowledge and nested random structures. In this study, we apply these modeling strategies to a real-world dataset on AOD, in which repeated binary and continuous outcomes were measured for each subject. Our objective is not limited to the application; we emphasize evaluating different MCMC algorithms in estimating model parameters and selecting the most appropriate model structure for such data.

Repeated measures data are a type of longitudinal data in which repeated measurements (observations) of the same dependent or response variable are taken at two or more points in time, space, or across multiple occasions. Repeated measures are obtained by taking multiple measurements on the same experimental unit, such as an individual, animal, or machine [2]. Research on repeated-measurement data is increasing rapidly due to the high demand for advanced statistical techniques to analyze it across disciplines such as life sciences, public health, biostatistics and epidemiology, agriculture, Econometrics, Social Sciences, environmental studies, etc. [3].

Statistical modeling of repeated-measures data is complicated by two types of dependencies [4]: first, the dependency between the dependent and explanatory variables, and second, the dependency between the observed measurements of the dependent variables. In many fields of study, multivariate data are often generated with multiple outcomes. It is possible to combine continuous and discrete response variables. Due to their complexity, the data are often handled by applying regression analysis to each outcome independently, even when they are related. In studies with mixed data comprising both continuous and binary outcomes, multivariate techniques were more efficient than univariate approaches. In general, more accurate parameter estimates can be obtained by jointly estimating parameters using MCMC estimation methods in Bayesian inference. The dependent variables can be of two types. First is categorical (qualitative): ordinal or nominal, and second is quantitative: discrete or continuous. When analyzing repeated-measures multivariate data, the type of dependency between measurements arising from repeated events within the same participants is of great concern. In other words, the nature of the correlation within and between individuals must be taken into account [4–6].

Hierarchical statistical models, also called random effects, mixed effects, or multilevel models, are widely used for data that are structured into groups of units. By examining repeated-measures data as a two-level hierarchy, researchers can build more complex hierarchical model structures [7]. Bayesian statistical techniques provide flexible modeling assumptions that enable models to accurately capture the complexity of real-world data. The dependent variable's distribution, probability functions or prior distributions, regression structure, numerous layers of observational

units, etc., are some examples of these adaptable modeling tools. A collection of estimating techniques known as MCMC approaches is used to fit realistic, intricate functions of parameters with subtle variations to Bayesian models, which are often compared with more conventional models [8].

According to research by Lages and Scheel (2016), Bayesian logistic mixed models with latent imputation and group-specific parameters better accounted for the data [9]. At the same time, the classical logistic hierarchical model with mixed effects also performed well. Different MCMC approaches vary in their performance in terms of speed and convergence, depending on the model structure [10]. MCMC approaches are presented both theoretically and practically using the R, BUGS, and STAN packages. Lemoine (2019) used R simulations to show how informative priors affect posterior parameter estimates and model performance, which explains the hierarchical modeling of the random effects structure and its impact on statistical inference [11]. However, computational problems may increase as the number of outcomes increases, leading to a strong dependency structure [12].

A Bayesian probabilistic technique is proposed to analyze regression models with mixed response or dependent variables as continuous and categorical variables, by incorporating prior distributions. Among the important aspects of multivariate models for mixed-type data is the investigation of the correlation structure of dependent-variable vectors to answer multivariate queries [13–15]. It provides control over the rise in type I error that occurs when univariate analyses are performed across different dependent variables. Multiple comparisons of model parameters are performed without accounting for random terms. Moreover, because different sets of individuals contribute to each analysis, it provides more precise parameter estimates by accounting for relationships among dependent variables and facilitating interpretation in the absence of dependent variables [16].

Tidemann-Miller et al. (2016) proposed a Bayesian approach for jointly analyzing multiple functional dependencies of different types (e.g., binary/categorical and continuous data) [17]. In this case, it models the dependence between functional responses via the multivariate latent normal process and the dependence of the latent process. There is a need to jointly analyze multivariate mixed outcomes while accounting for explanatory variable effects and considering all variance-covariance structures [18]. Tate and Pituch (2007) demonstrated the usefulness of multivariate multiple outcomes in a hierarchical linear model from randomized field experiments and simulation data for a hypothetical scenario [19]. Analyzing multi-outcome longitudinal data in a linear hierarchical model provides great flexibility for modeling between- and within-group correlation in multi-outcome repeated measures. A robust Bayesian approach to the multivariate Student-t linear mixed model can be equipped with a computationally efficient inverse-Bayesian-formula (IBF) sampler, along with a Gibbs sampler [20].

According to Alfo and Giordani (2022), a flexible regression model for multivariate dependent variables with mixed distributions can be fitted by considering several (conditionally independent) univariate regression models with random effects specific to the outcome [21]. This multivariate model accounts for subtle heterogeneity specific to the outcomes and for dependence among them through their joint distributions. It also opens up more general dependence structures among random effects in regression Equations with different dependent variables.

The Bayesian Hierarchical Modeling (BHM) with mixed categorical (Binomial) and continuous (normal) dependent variables can be modeled on incomplete cross-sectional data. In this case, the model-dependent variable indicator for the categorical dependent variable can depend on the continuous dependent variable. Such a modeling phenomenon is demonstrated by data from an observational study, in which the effect of parents' psychological disorders on both verbal comprehension scores and the presence of negative symptoms in their children was modeled despite missing dependent variables [22].

Hierarchical models with multiple continuous dependent (outcome) variables, each with a Normal distribution, are common in applied research. However, it is less commonly applied, and it is possible to combine the analysis of different types of dependent variables in a hierarchical multivariate mixed dependent variables model; for example, we can include a continuous dependent variable, such as blood pressure, alongside a categorical dependent variable, such as smoking

status, in multivariate models [23]. In this study, a mixed-response modeling approach was applied to repeated-measures data to examine correlational effects.

This manuscript's main objective is to apply Metropolis-Hasting, Gibbs sampling, and HMC Bayesian MCMC methods for parameter estimation in hierarchical models of repeated-measures data with mixed dependent variable types, and to examine model selection methods.

## 2. Materials and methods

Most statistical models assume that observations are identically and independently distributed (iid), drawn from a single distribution, and that one or more unknown parameters are present. However, in many cases, it does not make sense to treat measurements as identically and independently distributed (iid) with the same parameter(s) from the same distribution [24]. For continuous variable (e.g., income, score, CD4 count, grade point average), and categorical (e.g., students' attendance status, patient health status, academic status) dependent variables, single-level classical linear and binary logistic or multinomial logistic regression models can be generalized to a Bayesian hierarchical/multilevel/ model by adding prior distributions and allowing the regression coefficients to vary randomly over the clusters.

### 2.1. Bayesian hierarchical mixed (binary and continuous) responses modeling

Modeling mixtures of Normal, Bernoulli (Binomial), and Multinomial distributed dependent variables in a repeated-measure data structure is not a simple task. This is because of differences in the distributional assumptions of the various link functions and exponential families for our different types of dependent variables [25]. Currently, using MCMC estimation approaches in R2MLwiN, we can only fit mixtures of Normal and Binomial dependent variables, and then only one probit link function for the Binomial. This is because only this combination of dependent variables results in a definable distribution for the lowest level of residuals. Other combinations of dependents can be handled by assuming independence at the lowest measurement level (often unrealistic) [26].

In the continuous response case, a linear hierarchical (mixed) model is typically assumed, in which the conditional response distribution is normal, $y_{ij}|\mu_{ij} \sim N\left(\mu_{ij}, \sigma_e^2\right)$, and the link function is the identity link $\mu_{ij} = \eta_{ij}$. The variance $V\left(\mu_{ij}\right) = 1$ and the random variation parameter is $\phi = \sigma^2$.

Binary responses are assumed to be independently Bernoulli distributed as $y_{ij}|\mu_{ij} \sim$ Bernoulli $\left(\mu_{ij}\right)$ for given $\mu_{ij}$. Here, the conditional expectation $\mu_{ij}$ is also the conditional probability $P\left(y_{ij} = 1\right)$ for given values of the covariates/ predictors/ and random effects. The logit link is the most widely used link function, expressed as

$$\eta_{ij} = logit\left(\mu_{ij}\right) = log\left(\frac{\mu_{ij}}{1 - \mu_{ij}}\right) = log\left\{\frac{P\left(y_{ij} = 1\right)}{P\left(y_{ij} = 0\right)}\right\}$$

(1)

The goal of the Bayesian paradigm is to calculate (or estimate) the joint posterior distribution, treating all unknown model parameters as random variables. In this study, we used a probit link function across all software implementations (R2MLwiN, MCMCglmm, and BRMS/Stan) to maintain comparability [27]. The general hierarchical bivariate mixed dependent variables model can be written as a more complex Bayesian general hierarchical linear model that can be expressed in matrix algebra as follows:

$$Y_{N\times 1} = \underbrace{X_{N\times p}\beta_{p\times 1}}_{\text{Fixed effects}} + \underbrace{Z_{N\times mq}U_{mq\times 1}}_{\text{Random effects}} + \underbrace{\varepsilon_{N\times 1}}_{\text{error term}}$$

(2)

where $\boldsymbol{X}\beta$ is the linear estimator of fixed effects: $\boldsymbol{ZU}$ is the linear estimator of random effects, and $\varepsilon$ is the residual term. In this idea, Bayesian analysis treats all fixed and random effects as random variables with their distributions. The multivariate (mixed) dependent variables can now be represented in matrix form, $Y_{n\times 1}$ is the vector of the dependent variable(s),

indicated as $Y_{it}^{(1)}$ as a continuous and $Y_{it}^{(2)}$ as a categorical response. The Bayesian hierarchical model considered here includes both categorical and continuous responses and includes both binary (Binomial) and continuous (normal) outcomes. Generalizing the model to multiple categorical (ordinal and nominal) response variables is straightforward in principle, since each categorical variable can be represented by a set of binary indicator variables.

Suppose $resp_1 = Y_{1sit} = (y_{1it}, \dots, y_{sit})' (t = 1, \dots, T_i = n_i)$ be a continuous response having $s$ measurement units. For these individuals, $Z_y = X$ is the vector of predictor/independent variables. Suppose $T$ numbers of observations on a categorical response, $resp_2 = Y_{2sit} = (y_{1it}, \dots, y_{Tit})'$. Thus, $Y_{it}^{(s)}$ is a response/ dependent variable for $i^{th} (i = 1, 2, \dots, N)$ individual subject with $t^{th} (t = 1, 2, \dots, n_i)$ observed measurements of $s^{th} (s = 1, 2, \dots, m)$ response/outcome type. Therefore, $Y_{it} = \left[ Y_{it}^{(1)}, Y_{it}^{(2)}, \dots, Y_{it}^{(s)} \right], resp_{1i}, resp_{2i}, \dots, resp_{mi}$ are $s$ distinct mixed outcome variables with a general case of normalized latent response $Y_{it}^{*(s)}$ or $resp^*{}_{si}$.

For a 2-level hierarchical model with measurement observations nested within individual persons, the multivariate mixed response/dependent/ variable model can be written as:

$$Y_{its} = \delta_i \left[ X_{it}'\beta_s + Z_{it}'\gamma_s + u_{is} \right] + (1 - \delta_i) \left[ X_{it}'\beta_s + Z_{it}'\gamma_s + u_{is} + e_{its} \right] \tag{3}$$

With a simultaneous response indicator, a dummy variable, $\delta_i = \begin{cases} 1, & \text{if } Y_{its} \text{ categorical} \\ 0, & \text{if } Y_{its} \text{ continuous} \end{cases}$.

In Bayesian statistics, three popular MCMC algorithms are used for estimating Bayesian marginal and joint posterior densities: Metropolis-Hasting, Gibbs sampling, and the new hybrid Monte Carlo algorithm. The HMC algorithm was implemented to update the parameters by suppressing random walk behavior and utilizing the gradient of the log-posterior density **[28]**.

**2.1.1. Specification of Models. Likelihoods:** The likelihood function is constructed by defining the conditional distributions of the observed dependent variable, given the model parameters, including the subject-specific random effects. A common approach for mixed responses is to use a latent-variable formulation for the binary outcome, which simplifies modeling the correlation between the two response types.

Let $Y_1 = Y_{1ij}$ be the continuous response variable and $Y_2 = Y_{2ij}$ be the binary response variable for individual subject $i, (i = 1, 2, \dots, N)$ at observed measurement $j, (j = 1, 2, \dots, n_i)$. For a vector of common predictor variables, $X_{ij}$, for both $Y_1$ and $Y_2$ response variables. Then,

The continuous response variable, $Y_1 = Y_{1ij}$, can be modeled with a linear hierarchical structure:

$$Y_{1ij}|\beta_{y_1}, X_{ij}, U_i, \sigma_{y_1}^2 \sim Normal\left(X_{ij}^T\beta_{y_1} + u_{i,y_1}, \sigma_{y_1}^2\right) \tag{4}$$

$$Y_1 = Y_{1ij} = X_{ij}^T\beta_{y_1} + Z_{ij}^T U_i + \epsilon_{1ij}$$

where, $X_{ij}^T\beta_{y_1}$ represents the fixed effects on $Y_{1ij}$ and $u_{i,y_1}$ is the individual subject-specific random effects, which accounts for the repeated measures in $Y_1$ within each subject. The error term, $\epsilon_{1ij} \sim Normal\left(0, \sigma_{y_1}^2\right)$, and $\sigma_{y_1}^2 = \sigma^2$ is the residual variance of the continuous response, $Y_1$. The parameter $\beta_{y_1}$ is the population-level (fixed effect) regression coefficient for $Y_1$. $U_i$ is the vector of subject-specific random effects.

To specify the likelihood (sub-model) for a binary response variable, $Y_2 = Y_{2ij}$, it is necessary to introduce a continuous latent variable, $Y_{2ij}^*$, that is linked to the binary response variable. A probit link is often preferred in a Bayesian context as it can simplify posterior sampling and is considered to have a fixed residual variance of 1 for identifiability; i.e., $\epsilon_{2ij} \sim Normal(0, 1)$.

$$Y_{2ij} = \begin{cases} 1 \text{ if } Y_{2ij}^* > 0 \\ 0 \text{ if } Y_{2ij}^* \leq 0 \end{cases}$$

Then, the latent variable can be modeled with a linear hierarchical structure:

$$Y_{2ij}^*|\beta_{y_2},\ X_{ij},\ U_i\ \sim Normal\left(X_{ij}^T\beta_{y_2}+u_{i,y_2},1\right) \tag{5}$$

$$Y_{2ij}^* = X_{ij}^T\beta_{y_2} + Z_{ij}^TU_i + \epsilon_{2ij}$$

where, $X_{ij}^T\beta_{y_2}$ represents the fixed effects on $Y_{2ij}$ and $u_{i,y_2}$ is the individual subject-specific random effects, which accounts for the repeated measures in $Y_2$ within each subject. $\beta_{y_2}$ is the population-level (fixed effects) regression coefficients for the binary response variable, $Y_2$.

The full likelihood for individual subject $i$ is the product of the likelihoods for each response type that requires integrating out the random effects:

$$L_i\left(y_i|\theta\right) = \int L\left(y_{1i}|U_i,\theta\right) \times L\left(y_{2i}|U_i,\theta\right) \times p\left(U_i|\psi\right) dU_i \tag{6}$$

where $\theta$ includes all fixed effects and variance components, and $\psi$ is the variance-covariance parameters of the random effects. The full random Bayesian model that combines the joint likelihood with prior distributions for all random and fixed effects is:

$$p\left(\theta, U|Y_1, Y_2\right) \propto p\left(Y_1, Y_2|U, \theta\right) \times p\left(U|\Sigma_u\right) \times p(\theta) \tag{7}$$

**Prior distributions:** Bayesian inference requires specifying prior distributions for all model parameters. These priors can be informative (based on previous research) or non-informative (diffuse), and they can vary depending on the model formulation and the R package used [29,30]. For the fixed effect parameters, $\beta_{y_1}, \beta_{y_2}$, the weakly informative (such as Normal (0,1) or flat priors are common choices among most R packages.

$$\beta_{y_1} \sim Normal\left(0, \Sigma_{\beta_{y_1}}\right)$$

$$\beta_{y_2} \sim Normal\left(0, \Sigma_{\beta_{y_2}}\right)$$

where, $\Sigma_{\beta_{y_1}}$ and $\Sigma_{\beta_{y_2}}$ are diagonal matrices with larger variances.

The prior for the continuous response's residual variance, a common choice is an inverse-gamma distribution in MCMCglmm and R2MLwiN, and half-t or half-Cauchy distributions in BRMS. Thus, a weakly informative prior can be specified as:

$\sigma_{y_1}^2 \sim$ Inverse $-$ Gamma($a = 0.01, b = 0.01$) Or $\sigma_{y_1}^2 \sim$ Students_t(3, 0, 10).

The prior for the vector of random effects, $U_i = \left(u_{i,y_1},\ u_{i,y_2}\right)^T$ can be specified as a multivariate normal distribution with mean zero and a covariance matrix, $\Sigma_u$, which captures the correlation between the random intercepts for the two response variables. Thus,

$$U_i\ |\ \Sigma_u \sim Normal\left(0,\ \Sigma_u\right)$$

$$\Sigma_u = \begin{pmatrix} \sigma_{u,y_1}^2 & \rho\sigma_{u,y_1}\sigma_{u,y_2} \\ \rho\sigma_{u,y_1}\sigma_{u,y_2} & \sigma_{u,y_2}^2 \end{pmatrix}$$

where, $\sigma_{u,y_1}^2$ is the variance of the random intercept for the continuous response variable, $Y_1$. $\sigma_{u,y_2}^2$ is the variance of the random intercept for the binary response variable, $Y_1$; $\rho$ is the correlation between the random intercepts for $Y_1$ and $Y_2$. The common prior choices for the random effects covariance matrix, $\Sigma_u$, is an inverse-Wishart distribution in MCMCglmm and R2MLwiN, and Lewandowski-Kurowicka-Joe (LKJ) distribution in BRMS:

$$\Sigma_u \sim Inverse - Wishart(v, S)$$

$$\Sigma_u \sim LKJ(2)$$

where: $v = k + 1$ for $k$ number of random effects in the model, and the scale matrix $S$ is an identity matrix. The full posterior distribution, $p\left(\theta | data\right)$, can be formulated as:

$$p\left(\theta | data\right) \propto L\left(data | \theta\right) \times p(\theta)$$

where, $\theta = \left(\beta_{y_1}, \beta_{y_2}, \sigma_{y_1}^2, \Sigma_u\right)$ is the vector of all model parameters; $L\left(data | \theta\right)$ is the joint likelihood function, and $p(\theta)$ is the prior distribution for $\theta$.

## 2.2. Markov Chain Monte Carlo computational methods

Bayesian computation methods are employed when the posterior distribution cannot be obtained in closed form, requiring the application of alternative techniques to estimate or sample from it. The joint posterior distribution can be sampled using a class of computational techniques known as MCMC methods [31]. These techniques rely on constructing a Markov chain from the posterior distribution, which serves as the stationary distribution. Thus, samples from the joint posterior distribution are generated after multiple iterations by repeatedly sampling from this Markov chain.

Model formulation directly affects the effective sample size (ESS) through factors such as spatial and autocorrelation, distinct likelihoods, prior specification, and design matrix complexity. A model with substantial autocorrelation (e.g., Markov Chain Monte Carlo) may have an ESS lower than its actual sample size with default priors, since each sample depends on the one before it [32]. A lower ESS could likewise be found in a more complicated model with multiple associated parameters. Conversely, models with low autocorrelation and independent predictors will have an ESS closer to the actual sample size. Aim for an ESS of 1,000 or more for stable estimates and 200 or more for a bare minimum, while there are other guidelines. Emphasize that greater values (400+) are essential for precise interval estimation, even though 200 is the absolute minimum for a very basic estimate [29,30].

In MCMC convergence diagnostics, the ESS measures the number of independent samples a MCMC sampler would need to produce the same precision as the current auto-correlated sample. The Metropolis-Hastings ESS is sensitive to the variance of the proposal distribution; high autocorrelation leads to a low ESS. The HMC and its variants generally have higher ESS than basic Metropolis-Hastings because their gradient-based updates enable more efficient exploration of the parameter space. However, it is still affected by adaptation, tuning, and the model's specific formulation. Gibbs samplers can achieve high ESS if the conditional distributions are easy to sample from and the variables are not highly correlated. Different MCMC samplers yield different ESS values because of their distinct mechanisms for exploring the posterior distribution. HMC generally yields higher ESS per iteration in many complex problems, whereas Metropolis-Hastings and Gibbs samplers can be less efficient, especially in high-dimensional or highly correlated posteriors [33].

Although R-hat = 1.0 is the ideal value and indicates perfect convergence across all MCMC chains, R-hat statistic, or potential scale reduction factor, values < 1.1 indicate chain convergence, while values > 1.1 are indicative of less mixing and stationarity. For HMC and other MCMC methods, an R-hat value of 1.01 indicates excellent convergence. However, a value slightly higher than 1.01 might still be acceptable, with 1.1 being a standard threshold for early stages. R-hat values

significantly greater than 1.00 suggest that the Markov chains have not mixed well or converged to the target distribution, with the highest acceptable value often considered to be around 1.01 for robust convergence. The recommended threshold for an R-hat value of 1.01 or less is a modern standard used by probabilistic programming software such as Stan [34,35]. Others propose improvements to the traditional R-hat and recommend the ESS > 400 threshold for reliable diagnostics, which is the standard used in STAN [36].

MCMC techniques have been widely used to generate samples from high-dimensional, complex distributions. HMC is the most efficient one that uses a variety of Bayesian computational techniques [37]. The HMC method converges more quickly than conventional Metropolis-Hastings and Gibbs techniques and is most effective when approximating complex data-structure models [31]. Well-known MCMC approaches are used in complex parameter structures [38], but they exhibit poor performance and slow convergence. Similar to Gibbs sampling, HMC uses a random proposal distribution centered on the current parameter value. However, unlike the Gibbs algorithm, HMC does not rely on parameter sampling from a conditional posterior distribution. The HMC has two advantages over other MCMC methods. First, the samples have little or no autocorrelation. The other advantage is fast mix-in: the chain converges quickly to the distribution. Accordingly, it is the most effective method for continuous distributions, with low sample rejection and low (auto)correlation [39].

The HMC typically performs better for hierarchical models [40–42], but performance depends on the parameterization, marginalization, and data size. With a small dataset, discrepancies in the information criterion may be due to implementation differences.

The Metropolis algorithm, first proposed by Metropolis et al. (1953), is used when an analytical expression for the posterior distribution is not available [43]. As explained in Table 1, the algorithm requires a proposal distribution with a density function. It precedes one step at a time, based on simulations from this distribution, and accepts or rejects each step based on a Metropolis-Hastings ratio. Let $Q\left(\frac{\theta_t^*}{\theta_t}\right) P(.) d$ enote the target distribution (the posterior distribution) and $Q\left(\frac{\cdot}{\theta_t}\right)$ be the proposal from which the candidate point is sampled at iteration time.

The Metropolis approach, which chooses a symmetric proposal/offer distribution, $Q = Normal$, was historically found before Metropolis-Hastings. Sometimes we may need an asymmetric proposal distribution, $Q\left(\frac{\cdot}{\theta_t}\right)$, which has a different acceptance probability than the random walk Metropolis. Therefore, Tables 1 and 2 differ in how they calculate the acceptance probability. Here, a sample drawn from the conditional distribution for the new value is proposed as $\theta_0$ the new value, which is accepted with a certain probability $Q(./\theta_t)$. Otherwise, it remains negative. The Metropolis-Hastings method uses a proposal distribution to generate new approximations to the parameter ensemble.

Gibbs sampling [39] can be viewed as a specific instance of the Metropolis-Hastings method, in which the proposal distributions are the full conditional distributions of the model parameters (Table 3). This ensures that all ideas are automatically approved, since the Metropolis-Hastings algorithm's acceptance probability is always 1.

**Table 1. Algorithm 1: Metropolis Algorithm: Random Walk Metropolis.**

| |
|---|
| **Entry: starting point $\theta_0$, Qproposal distribution, and number of iterations , T;** |
| 1. $t = 0, 1, 2, \ldots, T$ for: |
| 2. $\theta^* \sim Q\left(./\theta_t\right)$, sample check that meets the condition;$Q\left(\theta_t/\theta^*\right) = Q\left(\theta^*/\theta_t\right)$ Calculate the probability of acceptance;$p = \alpha\left(\theta_t, \theta^*\right) = min\left\{1, \frac{P(\theta^*)}{P(\theta_t)}\right\}$ |
| 3. Sample a random variable;$U \sim Uniform(0, 1)$ |
| 4. If$U \leq p$ : |
| 5. $\theta_t \rightarrow \theta^*$ |
| 6. $\theta_t \rightarrow \theta_0$ (return to attract new candidates), |
| 7. $t\ i$ increase |
| Output: $\theta$ |

**Table 2. Algorithm 2: Metropolis-Hastings Algorithm (General).**

| |
|---|
| **Entry: Q initial proposal distribution and T update numbers with** $\theta_0$ **starting point; (when** $\theta_0$ **is given)** |
| 1. Iteration, t = 0, 1, 2, . . . , T |
| 2. Select sample $\theta_{t+1}^* \sim Q\left(\theta_t, \theta^*\right)$, Q without proposal distribution (but $\theta_t$ associated); |
| 3. Calculate acceptance probability; $p = \alpha\left(\theta_t,\ \theta_{t+1}^*\right) = min\left\{1, \frac{P(\theta_{t+1}^*)Q(\theta_{t+1}^*, \theta_t)}{P(\theta_t)Q(\theta_t,\ \theta_{t+1}^*)}\right\}$ |
| 4. Sample a random variable; $U \sim Uniform(0, 1)$ |
| 5. If $U \leq p,\ \theta_{t+1} \to \theta_{t+1}^*$ |
| 6. If not, $\theta_{t+1} \to \theta_t$ (return to take a new sample), increase t. |
| Output:$\theta$ |

**Table 3. Algorithm 3: Gibbs Sampling.**

| |
|---|
| **Entry: find a way to exploit conditional distributions, such as starting from a given point, if given, and inverse sampling, for** $\theta_0$. t = 0, . . . |
| 1. Draw: $\theta_{t+1}^{(1)} \sim Q\left(\frac{\theta_{t+1}^{(1)}}{\theta_t^{(2)}}...\theta_t^{(D)}\right)$ |
| 2. Draw: $\theta_{t+1}^{(2)} \sim Q\left(\frac{\theta_{t+1}^{(2)}}{\theta_{t+1}^{(1)}\theta_t^{(3)}}...\theta_t^{(D)}\right)$ |
| 3. Draw: $\theta_{t+1}^{(D)} \sim Q\left(\frac{\theta_{t+1}^{(D)}}{\theta_{t+1}^{(1)}}...\theta_{t+1}^{(D-1)}\right)$ |
| 4. Set; then the value is sampled from. $\theta_{t+1} = \left(\theta_{t+1}^{(1)}...\theta_{t+1}^{(D)}\right)Q\left(\frac{Y}{\theta_t}\right)' Y$ |
| Output: $\theta$ |

The HMC, also known as hybrid Monte Carlo, can suppress such random walk behavior when model parameters are continuous rather than discrete by employing a clever auxiliary-variable strategy that transforms the sampling problem from one target distribution to another [44]. By following a series of steps guided by first-order gradient information, the HMC method circumvents the random-walk behavior and the sensitivity to correlated parameters that plague conventional MCMC techniques. HMC requires appropriate parameter settings based on attributes and population structure to fit the BHM. Moreover, HMC outperformed Gibbs sampling on simulated data [39]. The HMC algorithm (Table 4) is based on the Hamiltonian (total energy), which calculates the trajectory for a time T and then obtains the final position, $t = 0, . . . , X_T = X_{n+1}$.

## 2.3. Application data set

Arterial occlusive disease data has two widely used intrusive techniques for classifying each leg as either healthy (0) or diseased (1): ultrasound imaging and reduced cuff pressure (RCP) measures. The variables can be expressed as *Status* : $Y_{ij}^{(1)}$ is the $i^{th}$ patient's health status from the measurement on the $j^{th}$ side of the leg. *Score* : $Y_{ij}^{(2)}$ is the $i^{th}$ patient's disease severity score measurement on the $j^{th}$ side of the leg. *Ultrasound* : $X_{1ij}$ is the $i^{th}$ patient's ultrasound image score measurement on the $j^{th}$ side of the leg. *RCP* : $X_{2ij}$ is the $i^{th}$ patient's RCP measurement on the $j^{th}$ side of the leg. Here, $i = 1, 2, 3, . . . , 16$ and $j = 1, 2, 3, 4$. Peripheral vessels are the vein branches that join the main vein to transport the contaminated blood from the arteries to the heart and the artery branches that emerge from the aorta and deliver clean blood to the arms, legs, brain, and organs [45].

Atherosclerosis, caused by decreased blood flow, is typically the cause of arterial occlusive disorders, which include occlusion or constriction of the arteries in the legs (often the arms). The data were obtained on disease severity, patient

**Table 4. Algorithm 4: HMC Algorithm.**

| |
|---|
| **Entry: starting point and a velocity distribution** $\theta_0$  $\theta_0 = X_0 q(v)$ |
| 1. Set n = 0, . . . |
| 2. Set the starting position as follows: $X(t = 0) = X_n$ |
| 3. Sample a new random initial velocity $v(t = 0) \sim q(v)$; |
| 4. Integrate the orbit numerically with the total energy for some time (use the Leapfrog method) $H(X, v) = U(X) + K(v) = -\log p(X) - \log T$ |
| 5. Compute the probability of acceptance $\alpha\left(X_{n+1}, X_n\right) = \min\left\{1, \frac{\exp\left[-H(X_{n+1}, v_{n+1})\right]}{\exp\left[-H(X_n, v_n)\right]}\right\}$ |
| 6. To arrange $X_{n+1} = X(t = T)$ |
| 7. Increase |
| Output: $\theta$ |

health status, and other factors at Broadgreen Hospital in Liverpool in 1988/89 [46]. This small dataset might not be a representative sample for clinical practice and inferences. However, we used it because the study emphasized methodological application across distinct MCMC approaches with no identical priors or likelihoods. The flat lining of healthy peripheral arteries encourages continuous blood flow and inhibits clotting. Although it can affect any artery, peripheral artery disease most frequently affects the legs [47]. Thus, in the data, the total number of measurements collected was 256 (16 patients, with four measurements on each of the two legs [left/right] and on each leg [upper/lower]). The dataset includes a categorical (binary) variable representing health status, as well as three distinct continuous variables: disease severity score, RCP, and ultrasound image measurement scores.

The patient's health status and disease severity scores were considered outcome variables, and the patient's RCP and ultrasound image measurement scores were independent variables/predictors. Of all patient features considered in the dataset, patient health status, ultrasound measurements, RCP measurements, and disease severity scores are included. The categorical (binary) response was the patient's health status, and the continuous outcome variable was the patient's disease severity scores. In contrast, the ultrasound imaging score and reduction cuff pressure measurements were explanatory variables expected to jointly predict the patient's health status and disease severity scores. The structural summary of the AOD dataset in long format is shown in Table 5.

The summary Table 5 of the AOD dataset clarifies that the study includes N = 16 individuals or patients; 2 legs (Left/Right) per patient; 4 measurements or repeats per leg side (Upper/Lower/occasions), totaling eight (8) measurements per patient and 256 observations. The dataset is fully observed (balanced) for the primary variables used in the analysis and has no missing values.

## 2.4. Ethical statements

As secondary data analysis from publicly available sources was used in this study, it is noted that the original study from which the data were obtained received ethical approval. The current analysis uses anonymized, de-identified data and does not allow for re-identification of participants. We have also made efforts to mitigate bias by establishing clear criteria for data inclusion and ensuring transparency in our methods, thereby promoting the reproducibility of our results.

## 3.  Results and discussion

### 3.1.  Descriptive findings on correlation and random effects

From Fig 1, the bivariate relationships were all positive. Patients' health status shows a strong correlation with ultrasound image and disease severity scores. In addition, patients' disease severity score has a moderate positive correlation with

**Table 5. The summary structure of the arterial occlusive disease dataset in long format.**

| Patient ID | Leg | Leg-side | Measurement occasions | Score(Y1) | Status(Y2) | Predictor variable, $X_{ij}, Z_i$ | | |
|---|---|---|---|---|---|---|---|---|
| | | | | | | Ultrasound: $X_{1ij}$ | RCP: $X_{2ij}$ | |
| 1 | Left | Upper | 1 | $y_{11}$ | $y_{21}$ | $x_{111}$ $x_{211}$ | | $z_1$ |
| 1 | Left | Lower | 1 | $\vdots$ | $\vdots$ | $\vdots$ $\vdots$ | | |
| 1 | Right | Upper | 1 | $y_{1j}$ | $y_{2j}$ | $x_{11j}$ $x_{21j}$ | | |
| 1 | Right | Lower | 1 | $\vdots$ | $\vdots$ | $\vdots$ $\vdots$ | | |
| 1 | Left | Upper | 2 | $y_{1n_1}$ | $y_{2n_1}$ | $x_{11n_1}$ $x_{21n_1}$ | | |
| 1 | Left | Lower | 2 | | | | | |
| 1 | Right | Upper | 2 | | | | | |
| 1 | Right | Lower | 2 | | | | | |
| .... | ... | | ... | ... | | ... | | $z_2$ |
| 2 | Left | Upper | 1 | | | | | |
| $i$ | 1 | | | $y_{i1}$ | | $x_{i11}$ $\cdots$ $x_{i1p}$ | | $z_i$ |
| | $\vdots$ | | | $\vdots$ | | $\vdots$ $\ddots$ $\vdots$ | | |
| | $j$ | | | $y_{ij}$ | | $x_{i1j}$ $\cdots$ $x_{ijp}$ | | |
| | $\vdots$ | | | $\vdots$ | | $\vdots$ $\ddots$ $\vdots$ | | |
| | $n_i$ | | | $y_{in_i}$ | | $x_{i1n_i}$ $\cdots$ $x_{in_ip}$ | | |
| $N = 16$ | Right | Lower | 4 | $y_{11}$ | | $x_{N11}$ $\cdots$ $x_{N1p}$ | | $z_N$ |
| | | | | $\vdots$ | | $\vdots$ $\ddots$ $\vdots$ | | |
| | | | | $y_{1j}$ | | $x_{Nj1}$ $\cdots$ $x_{Njp}$ | | |
| | | | | $\vdots$ | | $\vdots$ $\ddots$ $\vdots$ | | |
| | | | | $y_{1n_1}$ | | $x_{Nn_N1}$ $\cdots$ $x_{Nn_Np}$ | | |

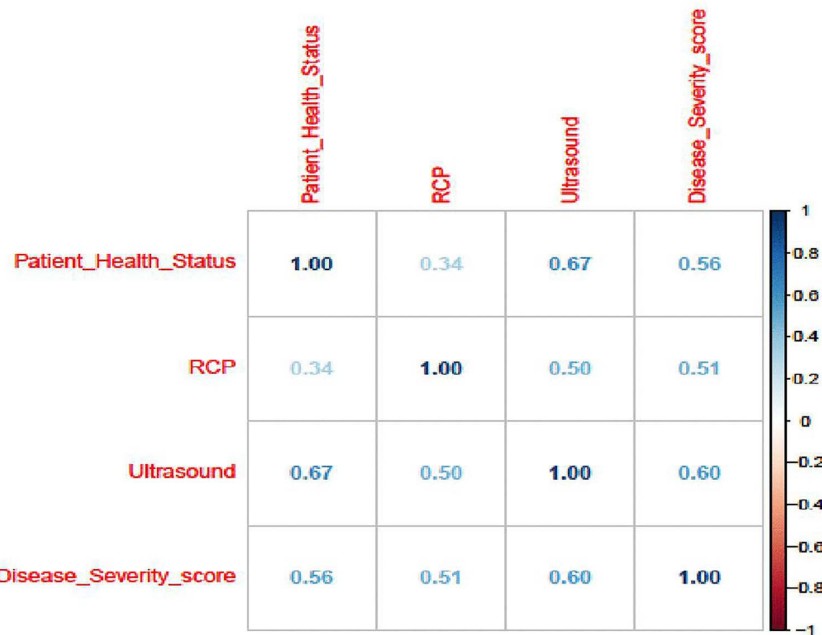

**Fig 1. Correlation analysis result of the study variables in the dataset.**

RCP and a strong correlation with ultrasound image score. The ultrasound image score shows a stronger correlation with both response variables than RCP.

The results in Fig 2 showed moderate variation in health status among patients, as reflected in their leg-side measurements. A large dispersion in disease severity scores was observed among patients based on leg-side measurements.

### 3.2. Fitted model results and Bayesian inferences

In Bayesian inference for joint modeling of patients' disease severity scores and health status, the null (empty) model, random-intercept model, and full random-slope models were fitted separately using three popular MCMC approaches. Thus, nine (9) distinct multivariate mixed response models were fitted, and their prediction performance was evaluated. These models were fitted using default likelihood and prior parameters, with the default settings of the R2MLwiN, MCMC-glmm, and HMC packages in R version 4.0.

In Table 6, the null (empty) model, the random-intercept model, and the full random-slope models were fitted separately using the Metropolis-Hastings approach for joint multivariate mixed outcomes. In the null model, both the intercepts for patients' disease severity score and health status were significant. The random part of the subject (level 2) was statistically significant. The random-intercept-with-fixed-slope model showed that both the intercepts for patients' disease severity score and health status (leg side only for patients' disease severity score) and the level-2 random variation were credible at the 5% level. The full random-intercepts-and-slopes model showed that the intercepts for patients' disease severity score and health status, the leg-side and ultrasound-image scores for joint outcomes, and the level-2 and subject variations by ultrasound were statistically significant at the 5% level.

In Table 7, the null Model, random-intercept model, and full random-slope model were fitted separately for joint multivariate prediction of disease severity score and patients' health status as mixed responses using the Gibbs sampling approach. Also, in Table 8, the null, random-intercept, and random-slope models were fitted separately for joint multivariate prediction of disease severity score and patients' health status using the HMC approach.

According to Table 8, posterior parameter (mean and SD) estimates from this best-fitting model via the Hamiltonian Monte Carlo method indicate that the predictors ultrasound and RCP were significantly associated with the patient's health status. A unit increase in ultrasound score has a significant impact (posterior mean of 12.5) on the health status of arterial occlusive disease. Ultrasound is a primary non-invasive tool for diagnosing, grading, and monitoring arterial occlusive disease, such as PAD. The results typically correlate highly with the extent and severity of the disease and are linked to cardiovascular risk. A unit increase in RCP score has a significant impact (posterior mean of 0.02) on health status outcome for patients with AOD. Moreover, there was significant variation in disease severity scores between patients (level 2) and significant variation in health status within patients (between measurements of the leg side). A random-effect SD indicates that disease progression trajectories vary between subjects, with a standard deviation of 0.12, representing the typical deviation of an individual's trajectory from the posterior mean in disease severity scores.

### 3.3. Model comparisons: Bayesian hierarchical multivariate mixed response

The three Bayesian hierarchical multivariate mixed models were assessed using the three MCMC methods across various information criteria, as shown in Table 9. According to Table 9, the full random-slopes hierarchical model has the lowest DIC, ICOMP, WAIC, LOO-IC, and 10-fold estimates across the Metropolis-Hastings, Gibbs sampler, and Hamiltonian Monte Carlo approximation techniques. Thus, among the above-established models, the last full random-intercepts-and-slopes model in the Hamiltonian Monte Carlo algorithm appears to be the best. Therefore, a full random intercepts and slopes model as a joint multivariate function of two responses and two explanatory variables (ultrasound and cuff pressure measurement), and the random coefficient for leg, ultrasound, and cuff pressure measurement has the best predictive performance for disease severity scores (Y1) and patients' health status (Y2). The consistency of an ensemble

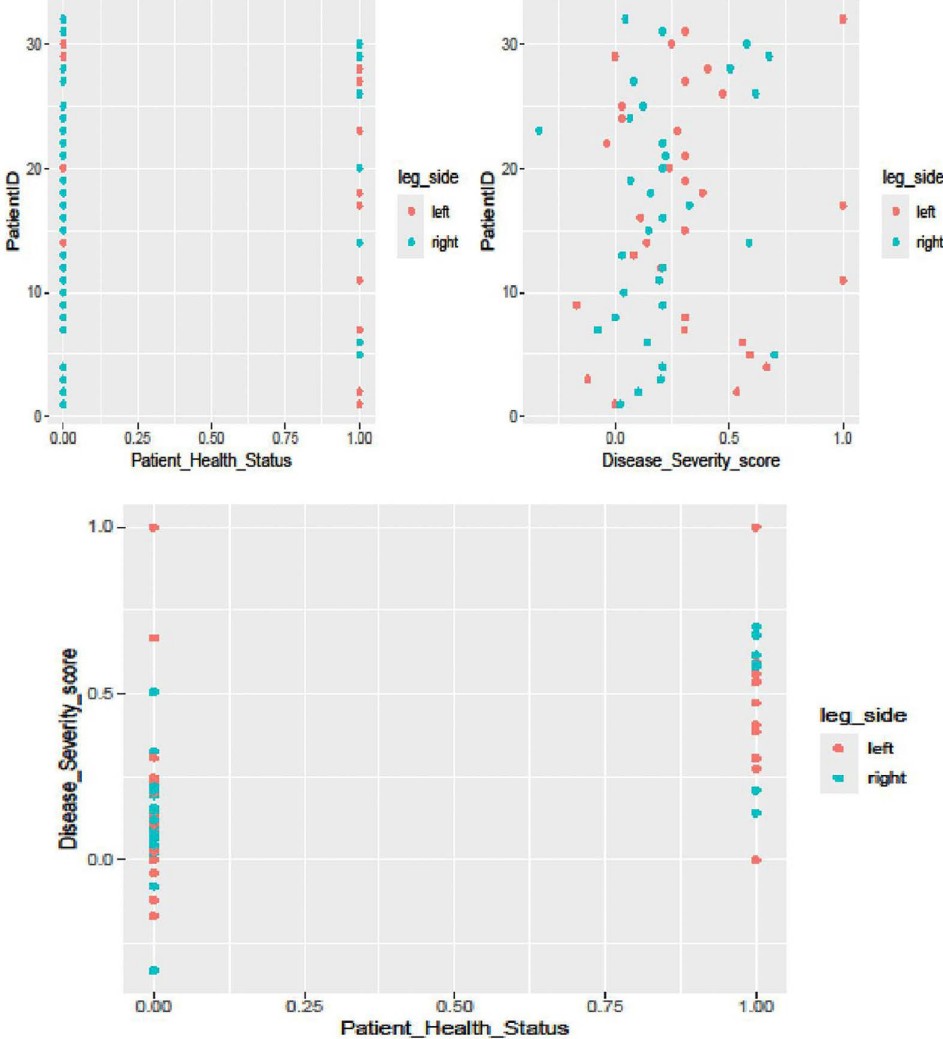

**Fig 2. Joint variation of patients' health status and disease severity score by leg side.**

of Markov chains was demonstrated by models with an ESS greater than 1000 and an R-hat value near 1.00, but not exceeding 1.10 [48]. Furthermore, the overall ESS of the models, and both the Bulk-ESS and Tail-ESS, must be at least 100 (approximately) per Markov chain to be reliable, and the corresponding posterior quantiles are reliable. These findings support the model convergence and consistency of MCMC chains.

The R-hat, Bulk-ESS, and Tail-ESS findings for the null, random-intercept, and random-coefficient models satisfied the convergence diagnostic metrics across all models using HMC. As a result, for stable estimates in each established model, the ESS and potential scale reduction (R-hat) convergence diagnostic measures are adequate [49]. The R2ML-wiN package employs a hybrid approach that can generate both frequentist and Bayesian-style output, serving as an interface to the standalone software MLwiN, which provides both estimation methods: Iterative Generalized Least Squares (IGLS) for maximum likelihood and MCMC for Bayesian inference. R2MLwiN offers the option to display the posterior mean and its corresponding posterior probability, but this relies on the assumption that the posterior distribution is approximately normal [50].

**Table 6. Fitted multivariate mixed response type model: Metropolis-hastings.**

| Parameter/ responses | Coefficient | Std. Err. | Posterior mean | Posterior probability | 95% CI Lower | Upper | ESS | Model type |
|---|---|---|---|---|---|---|---|---|
| | *Population-Level Effects: fixed effect parameters* | | | | | | | *Null Model* |
| Intercept. $Y_1$ | *0.3069* | *0.0532* | *5.77* | *0.000** | *0.2027* | *0.4111* | *1561* | |
| Intercept. $Y_2$ | *0.0210* | *0.0125* | *1.68* | *0.005** | *0.0124* | *0.0321* | *1120* | |
| | *Group-Level Effects: Random effect parameters* | | | | | | | |
| *Level 2* | *0.0904* | *0.2189* | *–* | *0.005 ** | *0.0082* | *0.8216* | *530* | |
| *Level 1* | *constant* | *1e-05* | *–* | *–* | *constant* | *1.000* | *1000* | |
| | *Population-Level Effects: fixed effect parameters* | | | | | | | *Random intercept with fixed slopes* |
| Intercept. $Y_1$ | *0.3115* | *0.0520* | *5.99* | *0.000** | *0.2410* | *0.3850* | *1011* | |
| Leg .$Y_1$ | *0.4312* | *0.2275* | *1.89* | *0.046** | *0.0041* | *1.0310* | *110* | |
| Ultrasound .$Y_1$ | *4.8906* | *1.0211* | *4.79* | *0.000** | *4.510* | *9.2540* | *208* | |
| RCP .$Y_1$ | *0.0051* | *0.0034* | *1.50* | *0.211* | *−0.0045* | *0.0198* | *650* | |
| Intercept. $Y_2$ | *0.0121* | *0.1151* | *0.105* | *0.454** | *0.0105* | *0.0456* | *450* | |
| Leg .$Y_2$ | *−0.0251* | *0.0121* | *2.074* | *0.002* | *−0.0157* | *0.0541* | *1000* | |
| Ultrasound .$Y_2$ | *2.751* | *0.321* | *8.57* | *0.546* | *−0.1251* | *11.45* | *800* | |
| RCP .$Y_2$ | *0.3954* | *0.1892* | *2.089* | *0.068* | *−0.3296* | *1.553* | *850* | |
| | *Group-Level Effects: Random effect parameters* | | | | | | | |
| *Level 2* | *0.0723* | *0.142* | *–* | *0.005** | *4e-04* | *0.4741* | *680* | |
| *Level 1* | *1.000* | *1e-05* | *–* | | *0.999* | *1.000* | *1000* | |
| | *Population-Level Effects: fixed effect parameters* | | | | | | | *Full random intercept and slopes* |
| Intercept. $Y_1$ | *0.1392* | *0.211* | *0.6597* | *0.0051** | *−0.0770* | *0.3293* | *1000* | |
| Leg .$Y_1$ | *−0.0234* | *0.634* | *0.0369* | *0.0012** | *0.1228* | *9.136* | *1000* | |
| Ultrasound .$Y_1$ | *0.2842* | *0.128* | *2.223* | *0.001** | *−0.6076* | *0.5474* | *1192* | |
| RCP .$Y_1$ | *0.0161* | *0.041* | *0.3961* | *0.2311* | *0.0017* | *0.0329* | *1000* | |
| Intercept. $Y_2$ | *−4.1972* | *0.8682* | *−4.83* | *0.0012** | *−5.9696* | *2.5298* | *1000* | |
| Leg .$Y_2$ | *0.43968* | *0.2275* | *1.93* | *0.0532** | *0.0068* | *0.9043* | *505* | |
| Ultrasound .$Y_2$ | *5.45584* | *1.3053* | *4.18* | *0.0231** | *3.1590* | *8.1262* | *890* | |
| RCP .$Y_2$* | *0.00675* | *0.0055* | *1.22* | *0.2226* | *−0.0040* | *0.0176* | *1000* | |
| | *Group-Level Effects: Random effect parameters* | | | | | | | |
| *Level 2** | *0.0482* | *0.0124* | *–* | *0.005** | *0.0339* | *0.0692* | *1000* | |
| $\sigma_{subj,\ ultrasound}$ | 5.425 | 0.133 | – | *0.0085** | 5.194 | 5.594 | 1000 | |
| *Level 1* | *0.9999* | *0.0032* | *–* | | *0.999* | *1.00* | *420* | |

* Parameters (HPD intervals) are credible at a 5% significance level. $Y_1$ = Disease severity scores *and* $Y_2$ = health status

MCMCglmm is a fully Bayesian package designed to fit Generalized Linear Mixed Models using MCMC methods. In a strict Bayesian framework, researchers typically use credible intervals (which represent the probability that the true parameter value lies within a given range) or posterior probabilities (e.g., the proportion of posterior samples that are above or below zero) to make inferences. Model fitting in Metropolis-Hastings is based on the random-walk behavior of algorithms 1 and 2, which can make it inefficient for exploring complex or high-dimensional distributions. In contrast, the HMC algorithm moves along trajectories using the posterior's gradient information, allowing it to take longer, more intentional steps into distant regions of the parameter space while retaining high acceptance rates. In this study, all continuous predictors were standardized (z-scored) before analysis to ensure coefficient comparability across algorithms. For the binary outcome (Y2), we utilized a probit link across all software implementations (R2MLwiN, MCMCglmm, and BRMS/Stan) to maintain

**Table 7. Fitted Multivariate Mixed Response Type Model: Gibbs Sampling.**

| Parameter/ responses | Estimate | Posterior probability | 95% CI | | ESS | Model type |
|---|---|---|---|---|---|---|
| | | | Lower limit | Upper limit | | |
| Population-Level (Location) Effects: fixed effect parameters | | | | | | Null Model |
| Intercept. $Y_1$ | 0.0825 | 0.032* | 0.0082 | 0.1541 | 2000 | |
| Intercept. $Y_2$ | -0.0731 | 0.604 | -0.3376 | 0.2042 | 1331 | |
| Group-Level Effects: Random effect parameters | | | | | | |
| Level 2: Intercept. | 0.040557 | 0.001* | 0.02238 | 0.0625 | 2000 | |
| Level 1: Intercept. | 0.359404 | 0.001* | 0.10301 | 0.7196 | 1000 | |
| Population-Level (Location) Effects: fixed effect parameters | | | | | | Random intercept with fixed slopes |
| Intercept. $Y_1$ | 0.1192 | 0.252 | -0.0770 | 0.3293 | 1000 | |
| Leg. $Y_1$ | -0.0224 | 0.682 | -0.1228 | 9.136 | 1000 | |
| Ultrasound. $Y_1$ | 0.2542 | 0.118 | -0.6076 | 0.5474 | 1192 | |
| RCP. $Y_1$ | 0.01804 | 0.044* | 0.0016 | 0.0349 | 1000 | |
| Intercept. $Y_2$ | -2.818 | 0.001* | -5.170 | -4.853 | 757.3 | |
| Leg. $Y_2$ | 2.759 | 0.132 | -8.807 | 7.727 | 1000 | |
| Ultrasound. $Y_2$ | 3.670 | 0.001* | 5.389 | 6.797 | 554.6 | |
| RCP. $Y_2$ | 0.4705 | 0.198 | -0.3296 | 1.553 | 868.5 | |
| Group-Level Effects: Random effect parameters | | | | | | |
| Level 2: Intercept ($Y_1$, $Y_2$) | 0.1594 | 0.001* | 0.07292 | 0.2618 | 991 | |
| Level 1: Intercept ($Y_1$, $Y_2$) | 2.400 | | -1.342 | 3.353 | 1000 | |
| Population-Level (Location) Effects: fixed effect parameters | | | | | | Full random intercept and slopes |
| Intercept. $Y_1$ | 0.1379 | 0.318 | -0.127 | 0.4270 | 1000 | |
| Leg. $Y_1$ | -0.0324 | 0.423 | -0.143 | 7.112 | 1000 | |
| Ultrasound. $Y_1$ | 0.1109 | 0.712 | -0.3666 | 0.6086 | 1000 | |
| RCP. $Y_1$ | 0.04212 | 0.948 | -0.0872 | 0.1048 | 1000 | |
| Intercept. $Y_2$ | -2.663 | 0.001* | -4.164 | -1.268 | 614.1 | |
| Leg. $Y_2$ | 42.089 | 0.120 | -14.88 | 121.03 | 1000 | |
| Ultrasound. $Y_2$ | 4.676 | 0.001* | 2.181 | 7.787 | 512 | |
| RCP. $Y_2$ | 0.3552 | 0.622 | -0.9402 | 1.610 | 816.8 | |
| Group-Level Effects: Random effect parameters | | | | | | |
| Level 2: Intercept. $Y_1$ | 0.3074 | 0.001* | 0.1085 | 0.5632 | 1100 | |
| Level 2: Intercept. $Y_2$ | 6.562 | 0.0545 | -6.475 | 6.462 | 1232 | |
| $\sigma_{ultrasound}(Y_1, Y_2)$ | 0.126(1.06) | | 0.463(8.2) | 0.106(9.2) | 1000 | |
| $\sigma_{RCP}(Y_1, Y_2)$ | 0.005932 | | -0.7625 | 0.005597 | 1000 | |
| Level 1: Intercept. $Y_1$ | 0.09915 | 0.001* | 0.05054 | 0.1643 | 1000 | |
| Level 1: Intercept. $Y_2$ | -0.09121 | | -0.0523 | 7.385 | 850 | |

* Parameters (HPD intervals) are credible at a 5% significance level. $Y_1$ = Disease severity scores *and* $Y_2$ = health status

comparability. However, the observed significant differences in parameter estimates (coefficients) or intercepts in Table 6 and Table 8 were expected due to parameterization differences between (Metropolis-Hastings and HMC) packages, such as prior distributions and residual variance scaling. Metropolis-Hastings packages, such as R2MlwiN, frequently use the Inverse-Wishart distribution as a conjugate prior for covariance matrices. When variances are close to zero, this can be both informative and biased, resulting in overly optimistic coefficient estimates. HMC packages, such as

**Table 8. Fitted Multivariate Mixed Response Type Model: Hamiltonian Monte Carlo.**

| Parameter/ responses | Estimate | Std. errr | 95% CI | | Bulk_ESS | Tail_ESS | R-hate | Model type |
|---|---|---|---|---|---|---|---|---|
| | | | Lower | Upper | | | | |
| Population-Level Effects: fixed effect parameters | | | | | | | | Null model |
| Intercept. $Y_1$* | 0.31 | 0.06 | 0.19 | 0.42 | 19419 | 12476 | 1.00 | |
| Intercept. $Y_2$* | -1.21 | 0.50 | -2.23 | -0.27 | 22094 | 10972 | 1.00 | |
| Group-Level Effects: Random effect parameters | | | | | | | | |
| Level 2: Subject. $Y_1$* | 0.17 | 0.04 | 0.12 | 0.24 | 7140 | 9756 | 1.00 | |
| Level 1: Leg. $Y_2$* | 0.59 | 0.31 | 0.06 | 1.23 | 2928 | 1806 | 1.00 | |
| Cor($Y_1$, $Y_2$): $\rho_{y_1,y_2}$ | 0.62 | 0.34 | -0.36 | 0.99 | 1407 | 918 | 1.00 | |
| Population-Level Effects: fixed effect parameters | | | | | | | | Random intercept with fixed slopes |
| Intercept. $Y_1$* | 0.12 | 0.06 | 0.01 | 0.23 | 15235 | 12391 | 1.00 | |
| Leg. $Y_1$ | -0.03 | 0.02 | -0.06 | 0.02 | 16326 | 11692 | 1.00 | |
| Ultrasound. $Y_1$ * | 0.28 | 0.11 | 0.08 | 0.48 | 13898 | 11334 | 1.00 | |
| RCP. $Y_1$ * | 0.01 | 0.01 | 0.001 | 0.03 | 15445 | 12553 | 1.00 | |
| Intercept. $Y_2$* | -4.31 | 0.88 | -6.17 | -2.70 | 15065 | 10209 | 1.00 | |
| Leg. $Y_2$ * | 0.44 | 0.22 | 0.01 | 0.88 | 16661 | 11103 | 1.00 | |
| Group-Level Effects: Random effect parameters | | | | | | | | |
| Level 2: $Y_1$ * | 0.12 | 0.03 | 0.06 | 0.18 | 488 | 4108 | 1.00 | |
| Level 2: $Y_2$ * | 0.38 | 0.29 | 0.02 | 1.09 | 3775 | 5936 | 1.00 | |
| $\rho_{y_1,y_2}$ * | 0.25 | 0.52 | 0.22 | 0.88 | 1023 | 1886 | 1.00 | |
| Population-Level Effects: fixed effect parameters | | | | | | | | Full random intercept and slopes |
| Intercept. $Y_1$ | 0.12 | 0.08 | -0.05 | 0.28 | 1449 | 1121 | 1.00 | |
| Leg. $Y_1$ | 0.02 | 0.04 | -0.06 | 0.05 | 4511 | 4795 | 1.00 | |
| Ultrasound. $Y_1$ | -0.02 | 0.12 | -0.01 | 0.04 | 1014 | 1858 | 1.00 | |
| RCP. $Y_1$ | 0.02 | 0.03 | -0.02 | 0.04 | 1064 | 1245 | 1.00 | |
| Intercept. $Y_2$ * | -9.89 | 4.89 | -23.73 | -4.03 | 1006 | 1028 | 1.00 | |
| Leg. $Y_2$ | 1.12 | 0.97 | -0.16 | 3.77 | 1712 | 2081 | 1.00 | |
| Ultrasound. $Y_2$ * | 12.51 | 5.72 | 5.01 | 26.81 | 2910 | 2912 | 1.00 | |
| RCP. $Y_2$ * | 0.02 | 0.02 | -0.01 | 0.09 | 1977 | 2608 | 1.00 | |
| Group-Level Effects: Random effect parameters | | | | | | | | |
| Level 2: $Y_1$ * | 0.12 | 0.03 | 0.06 | 0.18 | 488 | 4108 | 1.00 | |
| Level 1: $Y_2$ * | 0.38 | 0.29 | 0.02 | 1.09 | 3775 | 5936 | 1.00 | |
| $\rho_{y_1,y_2}$ | 0.08 | 0.25 | -0.42 | 0.56 | 1112 | 2776 | 1.00 | |
| Level 1: Leg * | 0.92 | 0.63 | 0.08 | 2.48 | 3500 | 3660 | 1.00 | |
| Cor($Y_1$, $Y_2$): $\rho_{y_1,y_2}$ | 0.08 | 0.25 | -0.42 | 0.56 | 1347 | 2770 | 1.00 | |

∗ Parameters are credible at a 5% significance level. $Y_1$ = Disease severity scores *and* $Y_2$ = patient's health status.

BRMS-Stan, advocate using the LKJ prior for correlation matrices and separate priors for standard deviations. This decoupling enables far more flexible and precise correlation estimation, yielding distinct (often more reliable) coefficients than those from Inverse-Wishart models.

The PSIS k-diagnostics report shown in Fig 3 is a method for improving the stability of importance sampling estimates by smoothing the importance weights. The stability and reliability of importance sampling estimates, as examined by the Pareto shape parameter, are assessed using PSIS k-diagnostics for LOO, and are within an acceptable range for reliable

**Table 9. Comparison of predictive performance for multivariate mixed response models using MCMC methods across various information criteria.**

| MCMC approximation | Model Type | Information Criteria (R2MLwiN, MCMCglmm and R-BRMS) | | | | |
|---|---|---|---|---|---|---|
| | | DIC | ICOMP | WAIC | LOO_IC | K-fold |
| Metropolis Hasting using R2MLwiN | Null Model | 177.61 | 2182.32 | 196.45 | 197.44 | 189.23 |
| | *Random intercept with fixed slopes* | 131.82 | 2178.11 | 168.88 | 195.65 | 178.59 |
| | *Full Random intercepts and slopes* | 119.92 | 1822.69 | 163.65 | 176.45 | 167.14 |
| *Gibbs Sampling using* MCMCglmm | Null Model | 186.13 | 2108.55 | 194.43 | 194.54 | 184.41 |
| | *Random intercepts with fixed slopes* | 132.68 | 2125.73 | 164.78 | 163.88 | 181.63 |
| | *Full Random intercepts and slopes* | 122.52 | 1712.27 | 163.69 | 162.95 | 179.49 |
| Hamiltonian Monte Carlo using BRMS | Null Model | 169.12 | 68.595 | 171.91 | 172.19 | 172.93 |
| | *Random intercepts with fixed slopes* | 129.02 | 61.84 | 136.46 | 137.56 | 142.33 |
| | *Full Random intercepts and slopes* | 118.22 | 18.11 | 133.44 | 133.71 | 129.85 |

*Note:* Widely Applicable Bayesian information criterion (WAIC), Leave-one-out information criteria (LOO-IC), K-fold cross-validation (K = 10-Fold), Bayesian deviation criterion (DIC), Bozdagan's information complexity (ICOMP) criterion.

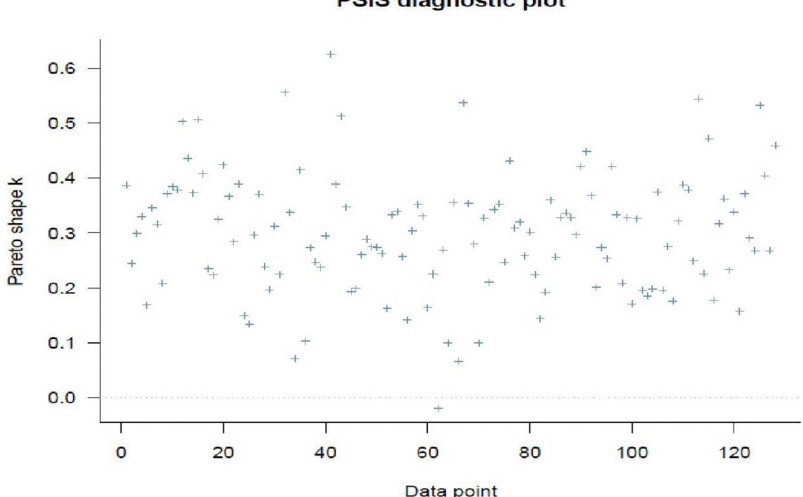

**Fig 3. PSIS k-diagnostics for LOO with full random intercepts and slopes model.**

estimates. The threshold is 0.7 in BRMS, and since the Pareto shape parameter k is not greater than or equal to 7, there are no problematic observations in the full random-slopes model.

A prior predictive check in Fig 4 is a crucial step in the Bayesian workflow used to evaluate if the chosen prior distributions for a model's parameters generate data that are consistent with existing domain knowledge or expectations, before considering the actual observed data. The prior predictive check diagnoses priors that are too strong, too weak, or poorly located, which could lead to problematic posteriors [51]. A prior predictive check is a statistical method in Bayesian analysis that evaluates whether a prior assumption is reasonable by simulating data from it before incorporating any actual data into the model [52]. These generate posterior parameter estimates and predicted values based only on the prior distributions of the model parameters. The weakly informative prior in the model, as shown in Fig 4(b), and its posterior lines are better overlaid compared to the default prior setups in Fig 4(a).

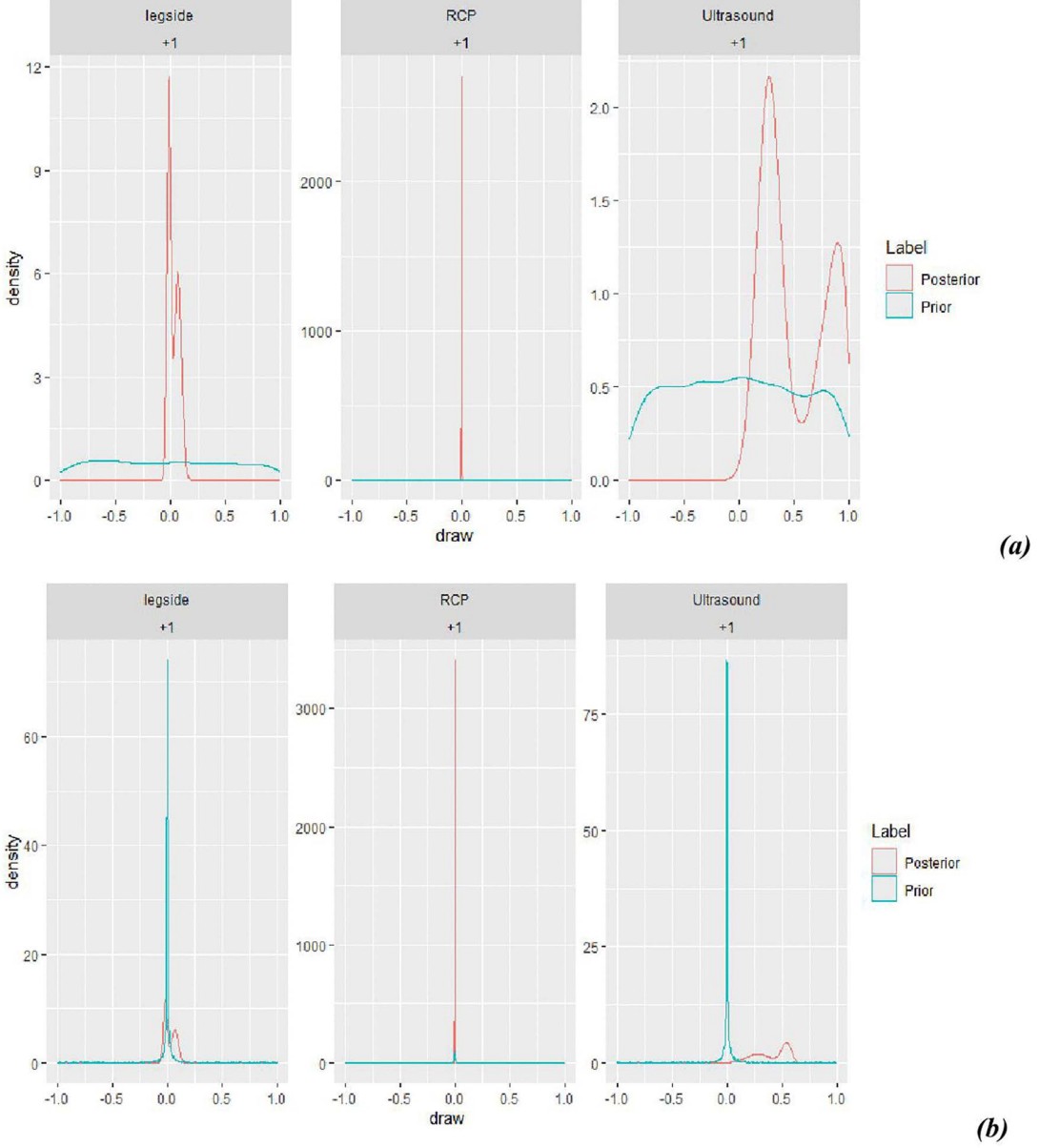

**Fig 4. Prior predictive check using default (a) and weakly informative (b) priors.**

In this application, hierarchical models, particularly the full random model in HMC, frequently mix better; however, performance depends on data size, marginalization, and parameterization. Because the dataset is so small, variations in the information requirements could arise across different implementations [53].

### 3.4. Evaluation of MCMC convergence diagnostics and conditional/marginal effects for the best-fitted model

The MCMC convergence diagnostics for ESS values in the fitted models under Metropolis-Hastings ensure that the posterior estimates are adequately characterized for the Null and Full random intercept and slope models. However, a slight discrepancy in posterior estimates was observed in the random-intercept-with-fixed-slopes model.

On the other hand, for all models fitted using Gibbs Sampling and HMC, the ESS were adequate (above 400) to ensure reliable posterior estimates and better convergence [36]. The ESS metric confirmed that the MCMC simulations ran for a sufficiently long time and mixed sufficiently well to produce reliable results via HMC. The ESS metric confirmed that the MCMC simulations ran for a sufficiently long time and mixed sufficiently well to produce reliable results via HMC.

The full random intercepts and slopes model estimated via HMC, with default for fixed terms and Half-Cauchy (0, 1) prior for scale parameters, is the best-fitted model in this application dataset. This choice yielded good model convergence and sufficient ESS values (i.e., greater than or equal to 1000). The Bayesian hierarchical random-slope model was fitted, and convergence diagnostics, effect directions, and marginal effects of predictors were checked. Additionally, a visually appealing posterior predictive check (PPC) plot of predicted versus observed data is presented.

Besides the necessary estimates of R-hat, the Bulk Effective Samples Size (Bulk-ESS) that indicate posterior means and medians were reliable, and the Tail Effective Samples Size (Tail-ESS) that indicate posterior variances and tail quantiles were reliable, the best convergences trace plots of the HMC approach are shown in Figs 5 and 6. Figs 7 and 8 show the marginal effects of each fixed and random term. Accordingly, ultrasound and leg type showed increasingly positive effects on patients' health status, but did not affect the disease severity score. Specifically, the marginal effects of ultrasound and RCP on patients' health status and disease severity score were positive and linear, as shown in Fig 8.

According to the posterior predictive check (PPC) in Fig 9, the random slope and intercept model fits well and produces nearly identical posterior observed density and posterior predicted scatter plots. The prediction plots show the dependent variables (patients' health status and disease severity score) of the Atherosclerotic diseases data set in the joint multivariate Bayesian hierarchical model. The PPC looks like the model is pretty good at retrofitting the data.

## 4. Conclusion

This study demonstrates the effectiveness of hierarchical Bayesian modeling for jointly analyzing multivariate repeated-measures data with mixed outcome types. Using clinical data on AOD, we compared the performance of three widely used MCMC methods—Metropolis-Hastings, Gibbs Sampling, and HMC—under various model specifications. The results revealed that the full random intercept and slope model estimated via HMC provides the best fit for this small clinical dataset, as measured by DIC, WAIC, LOO-IC, and ICOMP criteria, compared with other models. Despite inconsistent coefficients compared to other methods, this is a common phenomenon in small clinical datasets and cannot lead to generalization of the model to more hierarchical structures. However, this model-fitting and performance differences among MCMC samplers are often attributed to their distinct approaches to sampling from a probability distribution. These include differences in how the model is formulated (e.g., direct versus conditional sampling), the prior distributions used, and the specific decisions made regarding scaling and proposal parameters. When it comes to managing model complexity and the target distribution, each approach and package has its own unique advantages and disadvantages [54–56].

Clinically, the findings highlight the importance of jointly modeling patient health status (binary) and disease severity scores (continuous) to capture the correlation structure better and improve prediction. The significant association between ultrasound measurements and both outcomes supports its utility as a key non-invasive diagnostic indicator in peripheral arterial disease assessments. In addition, the best-fitted model showed that RCP and ultrasound image score had a significant effect on patients' health status.

In this specific bite-sized AOD dataset, with distinct prior and parameterizations, HMC produced significantly higher ESS and more stable R-hat diagnostics than Metropolis-Hastings and Gibbs sampling. However, we recognize these results may depend on the specific priors and parameterizations used and may not generalize to all small-sample hierarchical structures.

From a methodological perspective, our work underscores the value of gradient-based sampling methods, especially HMC, in estimating complex hierarchical models with high-dimensional parameter spaces. These models not only yield better fit and interpretability but also provide robust uncertainty quantification, critical in medical decision-making. The

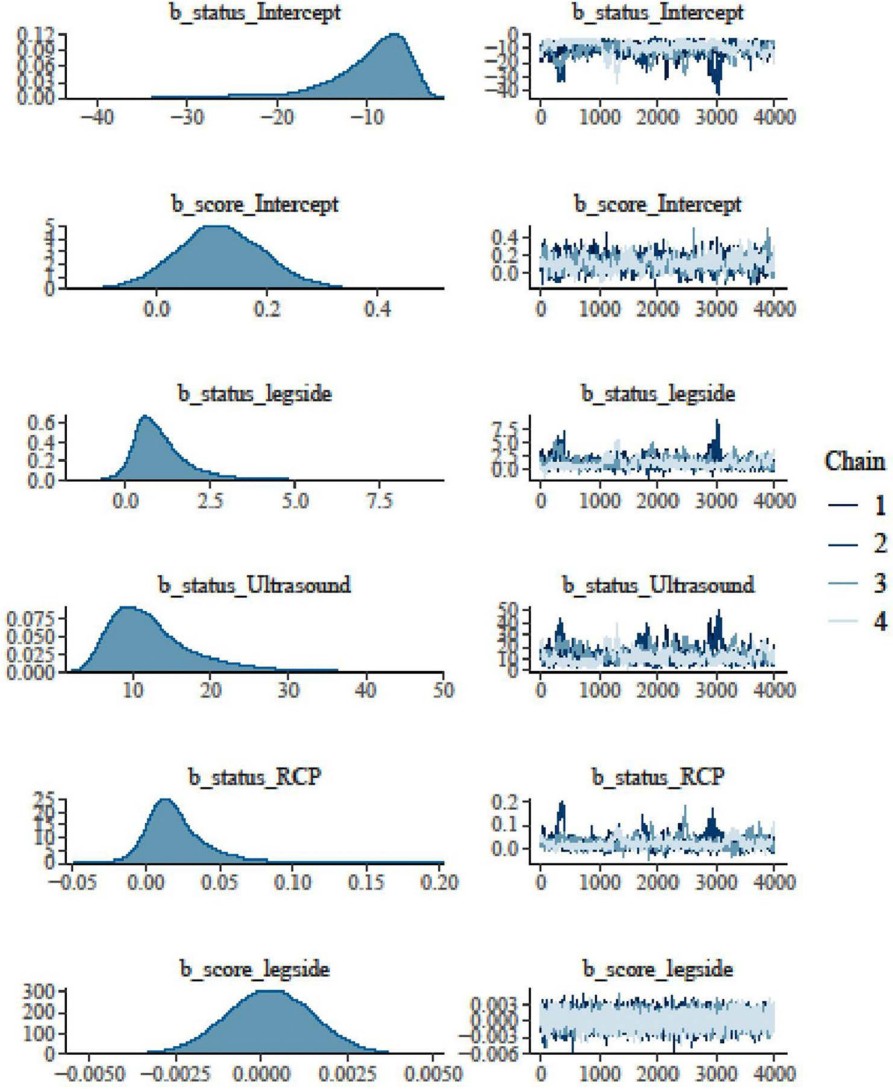

**Fig 5. Bayesian hierarchical random slope model convergence diagnosis.**

main contribution of this study is the methodological application of HMC to intricate hierarchical systems, despite using a historical dataset from 1988/89. The analysis demonstrates that when properly parameterized, HMC effectively navigates the high-dimensional geometry created by hierarchical correlation structures, which often cause convergence issues in standard Gibbs sampling and Metropolis-Hasting. Specifically, by using identical link functions, standardized predictors, and distinct parameterizations (default prior distribution settings), we observed that the random slopes were estimated with high efficiency. However, these methodological strengths must be balanced against two key caveats: prior sensitivity and sample size. The posterior distributions remain sensitive to the choice of hyper-priors, particularly when historical data are high-variance. While HMC is robust, the ESS for the late-period parameters suggests that results should be viewed as a proof-of-concept for the algorithm rather than a definitive clinical profile.

Future research should focus on applying this HMC framework to contemporary longitudinal datasets to validate these methodological lessons in a modern clinical context, and explore integrating more flexible priors (e.g., shrinkage or

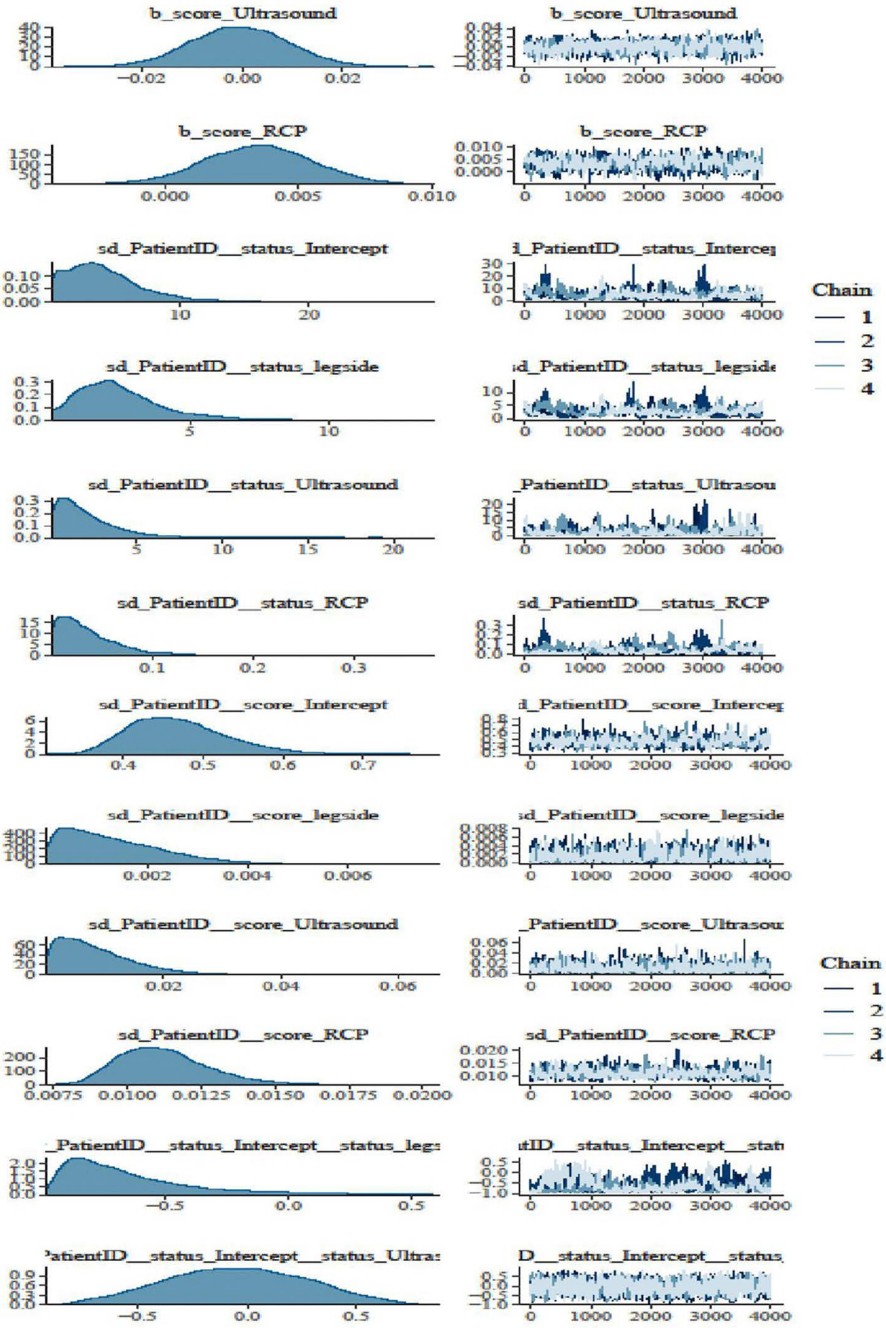

**Fig 6. Bayesian hierarchical random slope model convergence diagnosis.**

nonparametric priors), including time-varying covariates, and comparing with frequentist multilevel approaches. Moreover, extending this modeling framework to larger sizes and more recent clinical datasets can enhance its generalizability and translational potential in cardiovascular research.

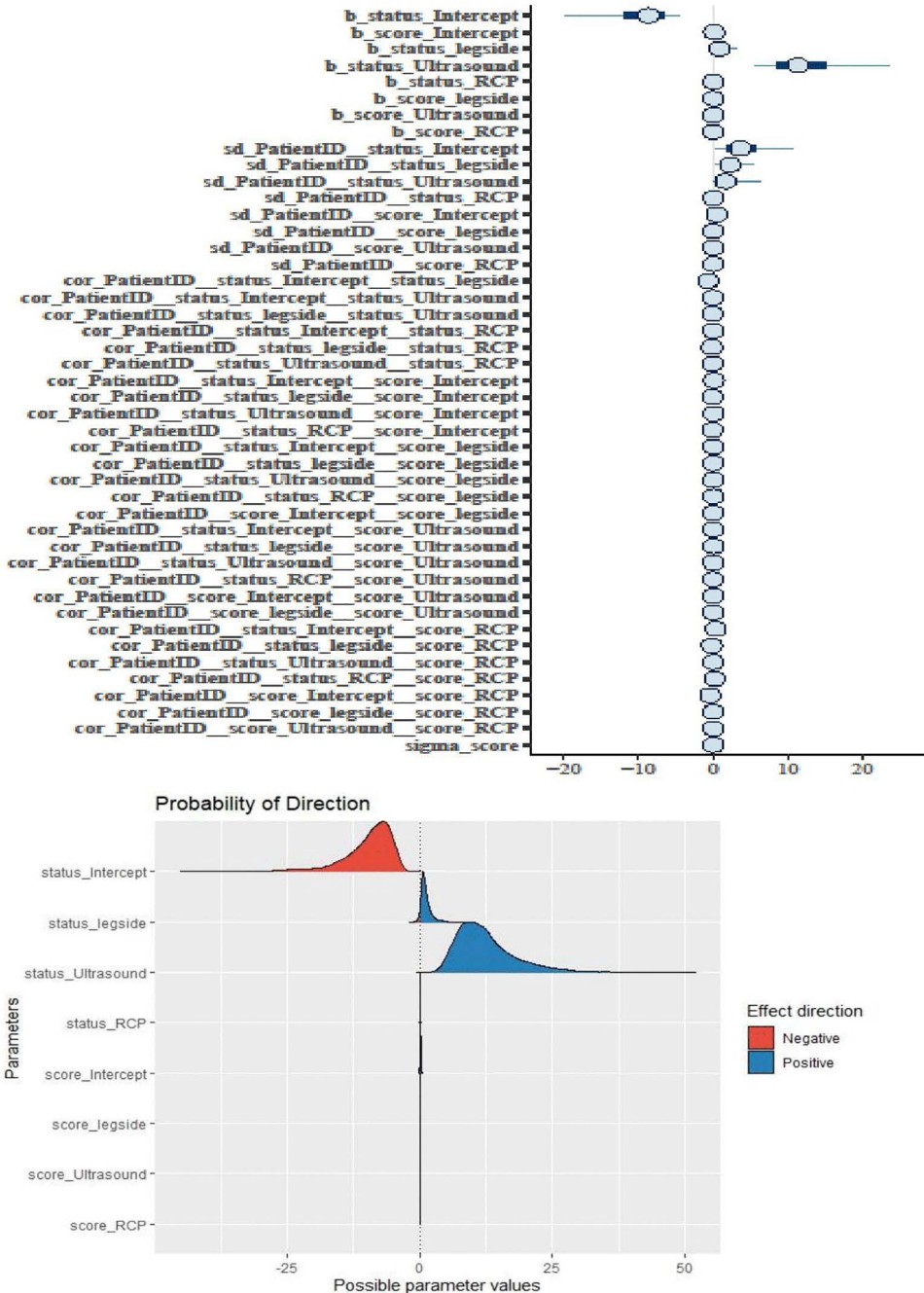

**Fig 7. Bayesian hierarchical random slope fitted model effects directions.**

## 5. Appendix

### 5.1. Posterior estimates (mean and median) and sensitivity analysis results

The choice of a prior depends on the data structure and software packages [57]. In R2MLwiN, we used the non-informative flat prior for the average fixed-effect parameter, and the scaled inverse Wishart (IW) distribution was

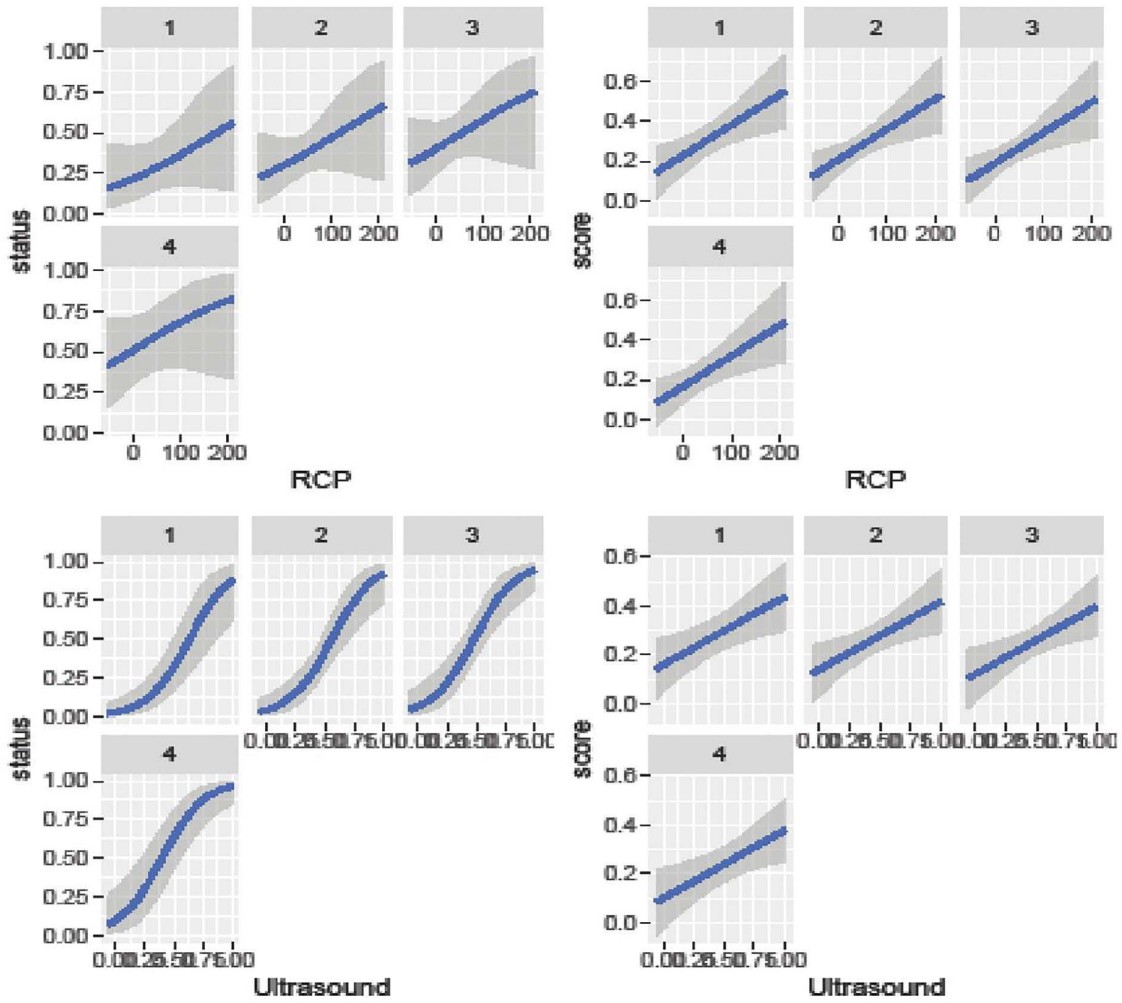

**Fig 8. Bayesian hierarchical random slope model marginal estimation effects.**

recommended for the random-effect variance–covariance matrices [58]. Priors beyond the informative and default were preferred for scale parameters [11].

The MCMCglmm library in R can perform BHM for continuous and mixed-dependent variables. The default priors for model coefficients and intercepts are uninformative, being standard normal distributions [59]. The prior distribution for the higher-order random effect is specified by two separate terms for the G-structure and the residual term R-structure, with the defaults V and nu (mean mu). A wide range of distributions and link functions is supported in BRMS [60]. By default, BRMS [61] uses a half Student-t prior distribution with 3 degrees of freedom. This prior is generally less informative, but leads to better model convergence than the half-Cauchy prior. Due to differences in model parameterizations and scaling, the predictive performance of models fitted with R2MlwiN, MCMCglmm, and RStan/BRMS was not identical. Different thinning, burn-in, and iteration values were applied to fit the models in the various packages. All models were run with four chains, each with four cores, a warm-up of 1000 iterations, a thinning interval of 1, and 10000 iterations using the Hamiltonian Monte Carlo approach.

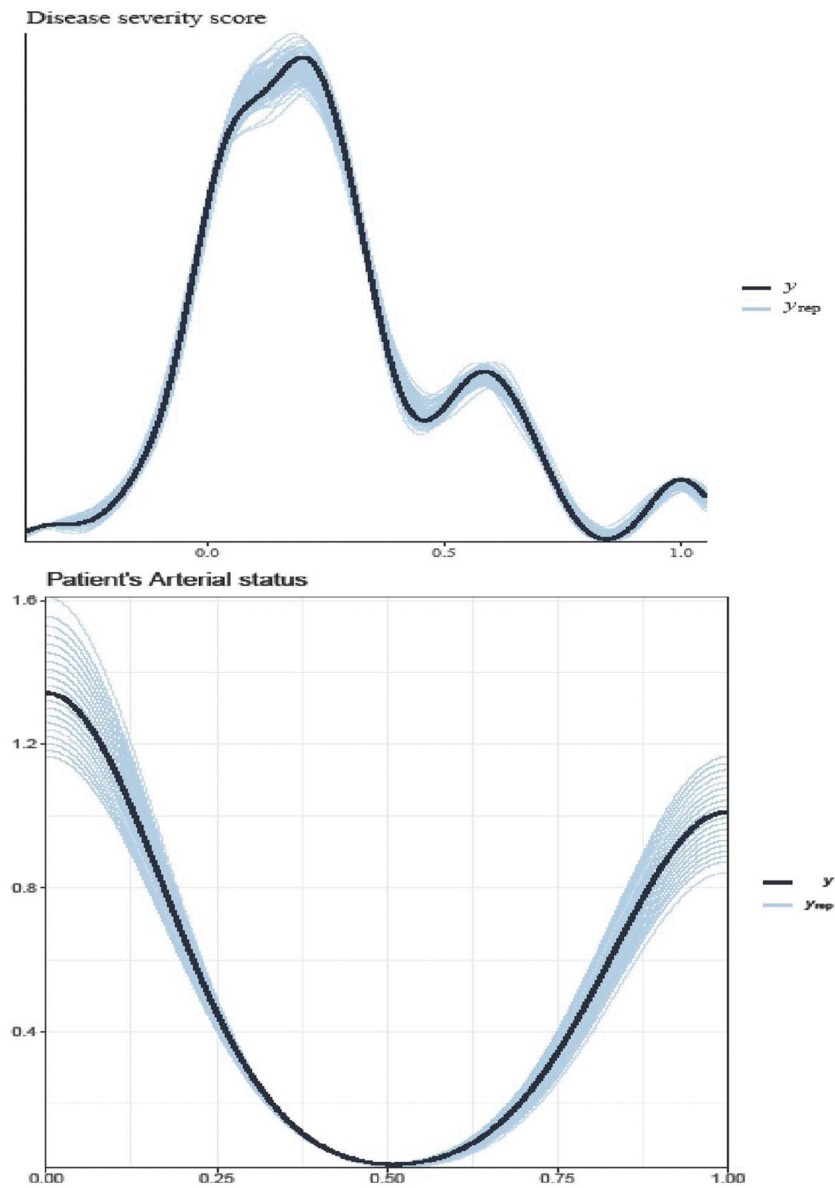

**Fig 9. Bayesian hierarchical random slope fitted model on observed (y) and predicted (y$_{rep}$) responses.**

Bayesian Models were fitted using the MCMCglmm package, employing Gibbs sampling for 5000 iterations after 100,000 burn-in iterations, with a thinning interval of 2000. Furthermore, the models with R2MlwiN for the Metropolis-Hastings approach were fitted for 10,000 iterations after 1,000 burns with a thinning interval of 10 as the default setup [62].

Different scholars have suggested various priors for components in hierarchical models of variability parameters, depending on the fitted Bayesian model structure and MCMC method [63–65]. Some researchers have proposed non-informative prior distributions, including uniform and inverse-gamma families, for use in Gibbs sampling [66]. Other researchers suggested a half-t family for the hierarchical model and demonstrated a relatively weakly informative prior distribution. Half-student-t prior, a default prior in BRMS for SD parameters, leads to better convergence. Still, the local

shrinkage parameters lead to more divergent transitions in the BRMS of Stan [67]. This study considered robust choices for group-level standard deviations in Bayesian hierarchical models, including half-normal (0, 5), half-Student t (3 degrees of freedom), and half-Cauchy prior distributions [68,69].

Kallioinen et al. (2024) and Wesner & Pomeranz (2021) proposed a half-normal distribution for SD priors in BRMS. Choosing a truncated normal distribution is a good idea because the standard deviation cannot be negative [70] and [71]. However, a prior on the random-effect parameter with a long right tail is "conservative" because it yields large SD estimates.

Other scholars (Gelman & Hennig, 2016; Zwet & Gelman, 2022) proposed a half-Cauchy prior with a mode at zero and a large scale parameter [72,73]. This explains the half-Cauchy prior's restrictive nature in providing sufficient information for the small number of groupings inherent in the hierarchical structure of the data. To reduce the likelihood of unrealistically large SD estimates, the BRMS-Stan documentation suggested using a half-Cauchy prior, which automatically bounds the SD at 0. R-Stan renormalizes the distribution used so that the sum of the area between the bounds is 1. The half-Cauchy (0, 1) prior is a special case of the half-Student t distribution, where the degrees of freedom are parameterized in terms of the standard deviation (SD). It occupies a reasonable middle ground between different prior classes, performing well near the origin. It does not lead to drastic compromises in the estimates of the population-level (location) and group-level effects of the parameter space [74].

Sensitivity analysis of priors (S1 Table) demonstrates the robustness of the Bayesian analysis across different prior distributions. According to Aguilar and Bürkner (2022), for each SD component of the random parameters in hierarchical models, any prior distribution is practically well-defined on the non-negative real numbers only [75]. In this study, we used the default in BRMS, the truncated Student's t distribution with 3 degrees of freedom, as a reference prior. Since negative values do not apply to standard deviation, we also used a very informative truncated normal prior with a mean of 5 and a standard deviation of 0.01. A half-Cauchy prior was also employed in the sensitivity analysis of its impact on the Bayesian hierarchical models for the applied dataset [76].

Models with an ESS greater than 1000 and an R-hat value closest to 1.00 but not exceeding 1.10 demonstrated the consistency of an ensemble of Markov chains [48]. Moreover, both Bulk-ESS and Tail-ESS should be at least 100 (approximately) per Markov chain to be reliable, indicating that the estimates of the respective posterior quantiles are reliable. In this study, the R-hat, Bulk-ESS, and Tail-ESS results for the null (empty), random-intercept, and full random-slopes models met the convergence diagnostics. Therefore, the ESS and potential scale reduction (R-hat) convergence diagnostic metrics are sufficient for stable estimates in each fitted model [32].

We conducted a sensitivity analysis of priors to scrutinize the results of the fully hierarchical specified model (random intercept and random slope using Hamiltonian Monte Carlo) based on the default (or reference) prior, using different prior distributions. The posterior mean estimates and 95% posterior density (HPD) intervals by the median for the fixed and random effects, including the SDs of the joint modeling disease severity scores (Y1) and health status (Y2), did not change much across the specified priors. This indicates a practically identical interpretation of the estimates, regardless of the priors. Since there was no significant difference in the percentage of models across the alternative prior II specifications, we reported the model results using the half-Cauchy prior in alternative prior I.

As explained by Depaoli et al. (2020), sensitivity analysis results can be presented visually, akin to Shiny app plots, or in a table format, indicating the degree of discrepancy in estimates or HPD intervals across parameters, as we present in the Appendix table below. We conducted a sensitivity analysis of the priors to examine the results of the final, fully random, hierarchically specified model in BRMS, based on the default prior (or reference), using two different prior distributions. The posterior distributions for the median and the 95% posterior density (HPD) as credible intervals (95% CI) for the fixed and random effects, including the SD in joint modeling of disease severity score (Y1) and health status (Y2), did not change much depending on the specified priors. This indicates a practically identical interpretation of the estimates, regardless of the priors. Therefore, we report the model results using the half-Cauchy prior. This approach provided good

model convergence and adequate ESS values (i.e., greater than or equal to 100) due to the lack of significant percentage deviation between models, regardless of the alternative prior specification.

Most primary inferences (S1 Table) regarding the covariate (Ultrasound) are robust to the choice of prior, as the signs and credible interval behaviors do not change meaningfully. However, the variance components and RCP effects are prior-sensitive; for these parameters, the posterior is more heavily influenced by the prior distribution than the data. Ultrasound (Y2) is highly robust, with a negligible −0.27% change under the Half-Cauchy prior and an 8.2% change under the Normal prior. RCP (Y1) shows high numerical sensitivity (−96.4% and −90.8% deviation), likely due to the small absolute magnitude of the estimate (0.0210 vs 0.5801). However, the practical inference remains similar as the effect size is near zero. The level 2 variance (Y1) parameter is highly sensitive to the choice of prior, with a −93.7% deviation under the Half-Cauchy prior and a −98.9% deviation under the Normal prior. This suggests that the data provide limited information about this specific variance component, making the prior choice influential. The correlation, $Cor(Y_1, Y_2)$: $\rho_{y_1,y_2}$), between Y1 and Y2 is sensitive to the scale prior, shifting from 0.0401 (default) to 0.0801 (+99.8% change) under the Half-Cauchy specification. However, it remained relatively stable under the Normal prior (−0.6%) and did not alter the primary conclusion that HMC provided better model fit.

## Supporting information

**S1 Table. Posterior estimates (mean and median) and sensitivity analysis results.**
(DOCX)

## Author contributions

**Conceptualization:** Endris Assen Ebrahim.

**Data curation:** Endris Assen Ebrahim.

**Formal analysis:** Endris Assen Ebrahim, Mehmet Ali CENGIZ.

**Investigation:** Endris Assen Ebrahim, Mehmet Ali CENGIZ.

**Methodology:** Endris Assen Ebrahim, Mehmet Ali CENGIZ.

**Project administration:** Endris Assen Ebrahim, Mehmet Ali CENGIZ.

**Resources:** Endris Assen Ebrahim, Mehmet Ali CENGIZ.

**Software:** Endris Assen Ebrahim.

**Validation:** Endris Assen Ebrahim, Mehmet Ali CENGIZ.

**Visualization:** Endris Assen Ebrahim, Mehmet Ali CENGIZ.

**Writing – original draft:** Endris Assen Ebrahim, Mehmet Ali CENGIZ.

**Writing – review & editing:** Endris Assen Ebrahim, Mehmet Ali CENGIZ.

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
