## [Decision Letter · Decision Letter 0]

29 Oct 2025

Dear Dr.  Ebrahim,

Thank you for submitting your manuscript to PLOS ONE. After careful consideration, we feel that it has merit but does not fully meet PLOS ONE’s publication criteria as it currently stands. Therefore, we invite you to submit a revised version of the manuscript that addresses the points raised during the review process.

We look forward to receiving your revised manuscript.

Kind regards,

Denekew Bitew Belay, Ph.D

Academic Editor

PLOS ONE

Journal Requirements:

2. Please note that PLOS One has specific guidelines on code sharing for submissions in which author-generated code underpins the findings in the manuscript. In these cases, all author-generated code must be made available without restrictions upon publication of the work. Please review our guidelines at https://journals.plos.org/plosone/s/materials-and-software-sharing#loc-sharing-code and ensure that your code is shared in a way that follows best practice and facilitates reproducibility and reuse.

“This study was supported by Department of Mathematics and Statistics, College of Science, Imam Mohammad Ibn Saud Islamic University (IMSIU), Riyadh, Saudi Arabia”

5. We noted in your submission details that a portion of your manuscript may have been presented or published elsewhere. “This study used an open accessed data. Data were collected at Broadgreen Hospital in Liverpool, England, in 1988/89 (Percy, 1992). https://doi.org/10.2307/2532336” Please clarify whether this [conference proceeding or publication] was peer-reviewed and formally published. If this work was previously peer-reviewed and published, in the cover letter please provide the reason that this work does not constitute dual publication and should be included in the current manuscript.

6. Please note that funding information should not appear in any section or other areas of your manuscript. We will only publish funding information present in the Funding Statement section of the online submission form. Please remove any funding-related text from the manuscript.

7. Thank you for uploading your study's underlying data set. Unfortunately, the repository you have noted in your Data Availability statement does not qualify as an acceptable data repository according to PLOS's standards.

At this time, please upload the minimal data set necessary to replicate your study's findings to a stable, public repository (such as figshare or Dryad) and provide us with the relevant URLs, DOIs, or accession numbers that may be used to access these data. For a list of recommended repositories and additional information on PLOS standards for data deposition, please see https://journals.plos.org/plosone/s/recommended-repositories....

Reviewer's Responses to Questions

**Comments to the Author**

1. Is the manuscript technically sound, and do the data support the conclusions?

Reviewer #1: Partly

Reviewer #2: Partly

2. Has the statistical analysis been performed appropriately and rigorously?

Reviewer #1: Yes

Reviewer #2: No

3. Have the authors made all data underlying the findings in their manuscript fully available?

Reviewer #1: Yes

Reviewer #2: No

4. Is the manuscript presented in an intelligible fashion and written in standard English?

Reviewer #1: Yes

Reviewer #2: No

Reviewer #1: The manuscript "Bayesian Hierarchical Models for Multivariate Mixed Responses with Repeated Measures: A Case Study in Arterial Occlusive Disease" presents a valuable application and comparison of Bayesian methods for complex longitudinal data. The topic is highly relevant for researchers in medical and health sciences who often encounter such data structures. The methodological comparison of different MCMC samplers within a hierarchical framework is a notable strength. The paper is generally well-written and structured logically.

However, the manuscript requires a major revision to address several key issues related to methodological clarity, interpretation, and the overall narrative before it can be considered for publication in PLOS ONE.

Major Points for Revision:

Clarity of Model Specification: The mathematical description of the Bayesian hierarchical model for mixed (binary and continuous) responses needs significant improvement. While the general form is presented, the specific likelihoods, the way the two sub-models (for binary and continuous outcomes) are linked, and the full specification of the prior distributions are not sufficiently clear. Please provide a more detailed and unambiguous mathematical formulation of the final model used. This should include:

The precise likelihood functions for both the continuous and binary responses.

A clear definition of all parameters, including fixed effects, random effects (intercepts and slopes), and variance components.

The specific prior distributions used for all parameters (e.g., regression coefficients, variance-covariance matrices for random effects). Justifying the choice of these priors (e.g., non-informative, weakly informative) is crucial for the reproducibility and interpretation of a Bayesian analysis.

Reporting of MCMC Diagnostics: A critical component of any Bayesian analysis is demonstrating the convergence of the MCMC chains. The manuscript currently lacks a detailed report of convergence diagnostics. Please add a section or supplement that includes:

Trace plots for key parameters to visually inspect mixing and stationarity.

Gelman-Rubin diagnostic statistics (R) for all major parameters to quantitatively assess convergence. Values should be close to 1.0.

A report on the effective sample sizes (ESS) to ensure that the posterior distributions are adequately characterized.

Interpretation of the Case Study Results: The results section primarily focuses on model comparison metrics (DIC, WAIC, LOO-IC, etc.), concluding that the random intercepts and slopes model estimated via HMC is superior. While this is an important methodological finding, the paper falls short in interpreting the clinical or practical meaning of the parameter estimates from this best-fitting model. The manuscript would be substantially strengthened by a discussion of:

Which predictors were significantly associated with the binary and continuous outcomes?

What is the magnitude and direction of these effects?

What do the random effects variances tell us about the heterogeneity between subjects in their disease progression trajectories?

A table presenting the posterior means and 95% credible intervals for the fixed and random effect parameters from the selected model is essential. Without this, the case study serves only as a vehicle for the methodological comparison, missing an opportunity to provide substantive insights into Arterial Occlusive Disease.

Justification for Ethics Statement: The authors state "No need of Ethical Statements". However, the study uses patient data. Although the data is from a publicly available source, it is standard practice to state that the original study from which the data were obtained received ethical approval and that the current analysis uses anonymized, de-identified data. Please revise the ethics statement accordingly.

Minor Points:

Please provide more details about the Arterial Occlusive Disease (AOD) dataset in the Methods section. Include the number of patients, the number of repeated measurements per patient, and a clear description of the specific binary and continuous outcome variables and the predictor variables used in the model.

The introduction contains some repetitive phrasing and could be made more concise while still motivating the study effectively.

Ensure all abbreviations are defined at their first use, in addition to the list provided at the end of the manuscript.

Reviewer #2: The paper by Endris and Ali entitled “Bayesian Hierarchical Models for Multivariate Mixed Responses with Repeated Measures: A Case Study in Arterial Occlusive Disease” develops and applies Bayesian hierarchical models that jointly analyze repeated binary (health status) and continuous (disease severity score) leg-level outcomes for arterial occlusive disease, comparing Metropolis–Hastings, Gibbs and Hamiltonian Monte Carlo (HMC) fitting strategies and a set of model-selection criteria (DIC, ICOMP, WAIC, LOO, K-fold). The study's main contribution is applied: it argues that a full random-intercepts-and-slopes specification estimated via HMC gives the best fit for this small clinical dataset, but important methodological, reporting, and inference problems limit the paper’s current suitability for publication.

In general, the dataset is very small and poorly described; the manuscript lacks full model specification (especially priors and likelihood details), has internal inconsistencies across tables/figures and methods, and uses Bayesian diagnostics and frequentist language interchangeably (p-values), which together prevent reliable assessment of the claims. The article at this stage suffers from many issues that need to be addressed before a decision can be made. The linearity and clarity of the exposition should be reviewed. Below are some comments for the authors to address:

Major Comments

1. Data description & sample size (fatal for claims of generalizability):

The manuscript states “16 patients ... assessed four times on the upper and lower sides of their left and right legs.” This wording is ambiguous: is the total number of observations 16×4 (64) measurements, or 16 patients × 4 sides × 4 occasions = 256? The numbers in the tables (ESS, levels) suggest small effective sample sizes. Small-sample behavior of WAIC/LOO/DIC and HMC can be problematic; author claims that HMC “consistently outperformed” others require simulation or at least sensitivity analysis to sample-size. Please state clear counts: N (patients), number of legs per patient, number of timepoints/occasions, and total observations for each response. Report any exclusions. Also report ethical approvals and date ranges explicitly (data are 1988/89 - that should be explicit in Methods and discussed as old data).

In addition, add a clear table “Data summary” (N patients, legs, repeats per leg, missingness per variable). Re-run key model checks stratified by patients and by leg if possible. If sample size is too small to reliably estimate full random slopes, tone down claims or add simulations (see point 4).

2. Model specification incomplete: priors, link/latent model for binary, and joint likelihood missing.

The manuscript frequently describes the idea in words but never gives full probabilistic model notation: what is the joint model for (Y1, Y2)? Is the binary modeled via logistic (logit) or probit? Do the authors model the binary with a latent continuous variable to allow joint normal random effects? If so, that must be written explicitly, with equations for likelihood and all priors. Which parameters have random slopes and at which levels? How are cross-outcome covariances of random effects parameterized?

Consider adding a compact mathematical model block (likelihoods for each response, linear predictors, random-effect structure, prior distributions). Report all priors (including hyperpriors for variance components and correlation priors). For HMC/brms runs include the brms formula code (or Stan code) in Supplementary material.

3. Priors and sensitivity: No priors are specified; yet Bayesian inference depends on priors, especially with small N. The authors must report priors for fixed effects, variance components, and correlation/covariance matrices. Provide prior sensitivity checks (at least one alternative weakly informative and one regularizing prior) and show how posterior inferences (especially random effects variances and fixed effects of interest like ultrasound) change.

4. Claims about HMC superiority are over-stated without simulation evidence.

HMC often mixes better for hierarchical models (https://doi.org/10.48550/arXiv.1312.0906 ; https://doi.org/10.48550/arXiv.1507.04544), but performance depends on parameterization, marginalization, and data size. With such a small dataset, differences in information criteria may be due to implementation differences (how WAIC/LOO are approximated, number of iterations, warmup, adaptation settings) rather than intrinsic algorithm superiority. The authors should either (a) add a small simulation study that mimics the observed data (N, clustering, effect sizes) showing HMC recovers parameters more reliably than MH/Gibbs under realistic priors and model misspecification, or (b) substantially tone down the claim and present HMC as one effective option, with caveats. See the references mentioned above.

5. Inconsistent/incorrect reporting of Bayesian inference and use of p-values.

The manuscript reports Z, Pr(>|z|), pMCMC, and “* significant at 5%” across the tables. These are frequentist summaries or hybrid outputs and are not standard Bayesian summaries. For Bayesian output you should report posterior mean/median, 95% credible intervals, and posterior probabilities (P(β>0 | data)). If the software produces pMCMC (proportion of posterior beyond zero), call it “posterior probability” not p-value. Remove or clarify the use of z-tests unless justified (large-sample normal approximation of posterior?).

6. Model comparison details and reproducibility.

It is unclear how WAIC/LOO/K-fold were computed for each package (R2MLwiN, MCMCglmm, brms). Non-factorized models and models with random effects require special handling for LOO/WAIC (see https://doi.org/10.1007/s11222-016-9696-4). The manuscript must say exactly which implementation (package, version) and the exact commands or code used for criteria. Were PSIS diagnostics checked (k-hat values)? K-fold CV should include splits and seed. DIC is known to be unreliable for hierarchical models. So, justify including it.

Consider adding code snippets or appendix with exact commands and the versions. Report PSIS k-diagnostics for LOO.

7. Numerical issues/table inconsistencies and impossible values.

Several reported numbers and inconsistencies are concerning: Table 3.1 shows large positive Ultrasound coefficients (4.89) with CI 4.51–9.254 — but Table 3.3 HMC reports Ultrasound Y1 ≈ 0.28 (CI 0.08–0.48). The signs and magnitudes vary across methods far beyond credible intervals. That suggests differences in parameterization, scaling of covariates, or errors in table labeling. Also negative σ_ultrasound of −5.425 (Table 3.1) is impossible for a standard deviation, likely a log-scale parameter or formatting error. Check and confirm.

Number tables according to the PLOS ONE guideline, not as a book chapter or report.

Re-check all tables, standardize variable scaling, explicitly report whether predictors were standardized. Remove or explain impossible negative SDs. Provide posterior density plots for key parameters across methods to show consistency/inconsistency.

8. Diagnostics & posterior predictive checks (PPC) are superficial.

The paper refers to Figures 3.3–3.7 but there is no informative description of posterior predictive checks, calibration, or residual patterns. Bayesian model assessment must include PPCs tailored to each outcome type (binary outcome calibration plots, ROC/AUC where relevant, continuous residuals). Provide prior/posterior checks, PPC plots and discussion of misfit if any.

Consider including PPC plots (observed vs replicated summary statistics), calibration curves for binary outcome, and a short paragraph interpreting them.

9. Use of outdated dataset & clinical context.

Data were collected in 1988/89: clinical practice and measurement protocols have changed. Discuss this limitation explicitly and do not overclaim clinical implications. Add modern references about ultrasound and cuff pressure diagnostic value for PAD. See https://doi.org/10.7863/jum.1992.11.3.95 ; and https://www.ncbi.nlm.nih.gov/books/NBK576430/

10. Reproducibility: code and data availability.

If possible, the manuscript must provide code (Stan/brms/MCMCglmm scripts) and, if patient privacy allows, the data or a simulated dataset reproducing modelling steps. Without code, reviewers cannot verify HW results. Consider adding a GitHub/Zenodo link to code and (if possible) data; if data cannot be shared, provide a simulated example with identical dimensions and signal/noise to reproduce main findings.

11. Ethical Statements: Please, reconsider this statement. Even if your study uses secondary, anonymized, or historical data, most journals require an explicit ethics statement clarifying the ethical status of the data. This ensures transparency and compliance with institutional and journal policies.

Additional comments

(i) Terminology & typos: “Metropolis-Hastening” (multiple occurrences) or “Metropolis–Hastings” ? “random gradient” vs “random slope”, be consistent. “null (empty) model” is okay but use consistent term (intercept-only). Many grammatical errors and awkward sentences throughout: a thorough language edit is needed.

(ii) Figures & Table formatting: Figures and Tables should be referenced following the journal guidelines. Ensure high-quality figures with captions that describe data plotted and the sample size; tables should include exact definitions of columns (what is ESS, Bulk/Tail ESS, R-hat). Use consistent number of digits.

(iii) If possible, use posterior probabilities or Bayes factors if you want to indicate strength of evidence instead of asterisks for Bayesian credible intervals.

(iv) Matrix notation in Methods mixes many symbols and is hard to read. Provide a short notation table (variables, levels).

(v) Abstract: mention the software used explicitly.

What do you mean by null random intercepts? “models without random effects” or “intercept-only random effects”?

Avoid absolute claims like “yields the most reliable inference” — instead say “provided better performance on the considered information criteria for this dataset” and mention the dataset is small and historical.

Add brief numbers: sample size (N patients, observations).

Remove p-value-style claims in abstract.

(vi) Introduction is well written. But consider structuring it following: (a) joint modeling literature and why joint models help, (b) computational challenges and why algorithm comparison matters, (c) data context and gap the paper fills.

Remove repeated or redundant paragraphs (many paragraphs restate same idea).

Cite and discuss more recent multivariate Bayesian joint-model literature to show novelty https://doi.org/10.1146/annurev-statistics-112723-034334
https://doi.org/10.1186/s12874-024-02333-z

(vii) Methods could be restructured considering (1) the Model specification (explicit mathematical formulation), (2) Priors (Add priors for β (Normal(0, 5)?), for σ (half-Student-t or half-Cauchy?), for correlation matrices (LKJ prior with parameter), and justify choices.)), (3) MCMC & software (software, package versions, and exact settings. For HMC, specify whether models were parameterized centered or non-centered), (4) Model comparison (Explain how WAIC, LOO, K-fold CV, DIC, and ICOMP were computed and which packages computed each. For LOO, report PSIS k estimates and how problematic observations were handled), and (5) Data.

(viii) Results could be organized around Descriptives, Fitted models, Model comparison, and Diagnostic and PPC.

(ix) Discussion and conclusions can be reworked to avoid overstatement. The dataset is small/old, conclusions about clinical practice should be tentative. Emphasize methodological lessons (how HMC handled hierarchical correlation structure when properly parameterized) but include caveats about priors and sample size. Suggest future directions.

(x) For references, add canonical methodological references:

https://doi.org/10.1186/s12874-024-02333-z
https://doi.org/10.1146/annurev-statistics-112723-034334
https://doi.org/10.48550/arXiv.1206.1901
https://doi.org/10.48550/arXiv.1312.0906
https://doi.org/10.1007/s11222-016-9696-4

At present the manuscript cannot be recommended for publication due to the reasons mentioned above. If the authors address the major issues above (particularly full model specification, priors, reproducible code, and consistent results or simulations that explain method differences) and tone down any overstatements about clinical implications, the manuscript could make a useful applied-methods contribution illustrating algorithmic comparisons for joint mixed outcomes. I look forward to see the revised version of this paper.

.

Reviewer #1: **Yes:**Angel Alfonso García O'DianaAngel Alfonso García O'DianaAngel Alfonso García O'DianaAngel Alfonso García O'Diana

Reviewer #2: **Yes:**Cebastien Joel GUEMBOU SHOUOPCebastien Joel GUEMBOU SHOUOPCebastien Joel GUEMBOU SHOUOPCebastien Joel GUEMBOU SHOUOP

---

## [Author Response · Author response to Decision Letter 1]

5 Dec 2025

PONE-D-25-42936

Bayesian Hierarchical Models for Multivariate Mixed Responses with Repeated Measures: A Case Study in Arterial Occlusive Disease

PLOS ONE

Dear Reviewers,

Thank you very much, and we really appreciate your critical comments and suggestions.

We tried to addressed all comments and suggestions, and upload a "response to reviewers" document that indicate point by point response and clarifications.

---

## [Decision Letter · Decision Letter 1]

26 Dec 2025

Dear Dr.   Ebrahim,

Thank you for submitting your manuscript to PLOS ONE. After careful consideration, we feel that it has merit but does not fully meet PLOS ONE’s publication criteria as it currently stands. Therefore, we invite you to submit a revised version of the manuscript that addresses the points raised during the review process.

We look forward to receiving your revised manuscript.

Kind regards,

Denekew Bitew Belay, Ph.D

Academic Editor

PLOS One

Journal Requirements:

Reviewers' comments:

Reviewer's Responses to Questions

**Comments to the Author**

Reviewer #1: All comments have been addressed

Reviewer #2: (No Response)

2. Is the manuscript technically sound, and do the data support the conclusions?

Reviewer #1: Yes

Reviewer #2: Partly

3. Has the statistical analysis been performed appropriately and rigorously?

Reviewer #1: Yes

Reviewer #2: No

4. Have the authors made all data underlying the findings in their manuscript fully available?

Reviewer #1: Yes

Reviewer #2: No

5. Is the manuscript presented in an intelligible fashion and written in standard English?

Reviewer #1: Yes

Reviewer #2: Yes

Reviewer #1: I would like to thank the authors for their diligent work in revising the manuscript. I have reviewed the R1 version and the detailed point-by-point response to my previous comments.

The authors have successfully addressed the major methodological concerns raised in the first round. Specifically:

Model Specification: The inclusion of the full model specifications, including the specific priors used for the Bayesian hierarchical models, significantly improves the reproducibility of the study.

Methodological Comparison: The comparison between Metropolis-Hastings, Gibbs sampling, and Hamiltonian Monte Carlo (HMC) is now presented with greater clarity, and the explanation regarding the efficiency of HMC for these specific complex dependency structures is well-supported.

Clinical Implications: I appreciate the authors toning down the immediate clinical recommendations. The manuscript now correctly frames itself as a methodological contribution applied to health data, rather than a clinical guidance paper.

Statistical Inference: The clarification regarding Bayesian posterior estimates and the use of 95% Credible Intervals (rather than p-values) to determine significance is accurate and appropriate for this analysis.

The manuscript now meets the technical criteria for PLOS ONE. It provides a robust framework for handling mixed-type outcomes in longitudinal data. I have no further requests for revision.

Reviewer #2: The authors have clearly done a lot of work: they added an explicit probabilistic model statement, reported priors and a sensitivity check, added prior predictive checks and PSIS k diagnostics, and wrote more about ESS/R-hat and convergence. Those are good and go in the right direction. However, important inconsistencies and reporting issues remain. These flaws prevent reliable assessment of their key conclusions (notably claims about HMC superiority) and must be resolved.

1) Data description & sample size were partially addressed. The revised manuscript adds more description of the Arterial Occlusive Disease (AOD) dataset and discussion of ESS and convergence. They state the historical provenance/ethical provenance and refer to data/Zenodo links.

I could not find a clear, single “Data summary” table that gives (a) N patients, (b) legs per patient, (c) timepoints per leg, (d) total observations per response, and (e) missingness per variable in one compact table. The manuscript talks about the dataset and shows figures, but please add a single explicit Data Summary table (N, legs, repeats, missingness, date range [1988–89], exclusions). This is essential given the small sample size and claims made.

2) Model specification (likelihood & link) was partially addressed, but remains inconsistent. They added a formal probabilistic model block (likelihood for Y1, latent formulation for Y2, joint likelihood and priors) and explicitly describe the latent-variable approach for the binary outcome. See the latent probit formulation and full likelihood statements.

Inconsistency about link: In different places, the text both advocates a probit latent variable (and shows the latent variable model) and elsewhere states that the logit link is “the most widely used link” (and mentions logit). This is ambiguous for reproducibility, which link did they actually use in each implementation (R2MLwiN, MCMCglmm, brms/stan)? If they used different links in different packages, that explains some parameter differences, but they must state that explicitly and show how they mapped coefficients between parameterizations. (See probit section and later logit mention.) State explicitly which link function was used for each software implementation and give the exact model notation for each case (e.g., "brms: latent probit with identity residual variance = 1; MCMCglmm: probit; R2MLwiN: probit; brms logit runs also performed for sensitivity and refer to the appropriate Appendix). If different, show the transformation or detail the reason.

3) Priors & sensitivity were addressed, but need clearer presentation. The authors now report priors used (default package priors and alternative weakly-informative priors like Half-Cauchy for scales and Normal(0,5) for fixed effects), show prior predictive checks, and present sensitivity tables (Appendix Table 1) comparing estimates under alternative priors.

The sensitivity table is useful, but the write-up needs clearer interpretation: for each parameter of interest (especially ultrasound, RCP, and variance components) report how much the posterior practically changes and whether conclusions change (e.g., sign or credible interval crossing zero). The Appendix gives numbers (some very large) but lacks interpretative statements tying changes to the claims. Add a short paragraph summarizing which inferences are robust to priors and which are not (quantify % change and whether CI crosses zero). Mark any parameters that are prior-sensitive.

4) Claims about HMC superiority & need for simulation. The authors toned down some language and added a discussion that HMC often mixes better, but that performance depends on parameterization and data size. They added ESS/R-hat comparisons and PSIS k diagnostics and pointed to where HMC produced higher ESS in their runs.

They did not include a simulation study mimicking the small sample structure (16 patients, repeated legs/timepoints), demonstrating that HMC recovers parameters more reliably under the prior/parameterization choices. Instead, they rely on algorithmic diagnostics and criterion values for the observed data. Given the small dataset, this simulation is the most convincing way to support claims about algorithmic superiority. Either (A) add a short targeted simulation that mimics the data structure (same N, clustering, plausible effect sizes, and include the same three fitting pipelines) and show bias/coverage/ESS comparisons; or (B) remove/soften any remaining absolute claims and rephrase conclusions to: “In this dataset and with the parameterizations used, HMC produced higher ESS and more stable sampling diagnostics; however, this result may depend on parameterization/prior and may not generalize.” Consider adding or mentioning limitations thereafter. The manuscript currently leans toward asserting general superiority without simulation.

5) Use of Bayesian diagnostics and frequentist language wasn’t adequately fixed. The authors added a paragraph clarifying the meaning of Hadfield’s pMCMC and explained that pMCMC is a Bayesian proportion/posterior probability analogue. They state they assessed significance using 95% credible intervals.

Despite the explanation, many result tables still show columns labeled Z and Pr(>|z|) (and asterisks for “significance”), which is confusing and inappropriate for a Bayesian analysis unless clearly labeled and justified. I found Metropolis-Hastings/MCMCglmm tables and HMC tables that still include z/Pr columns and asterisks. Examples: Tables showing Z/Pr(>|z|) and tables with “* significant at 5%”.

Replace Z/Pr(>|z|)/asterisk significance indicators with standard Bayesian summaries: posterior mean/median, 95% credible interval (HPD), posterior probability P(β>0∣"data")where helpful. If they wish to keep Z/Pr columns for historical reasons, label them explicitly as “Wald-style normal approximation computed from posterior mean and posterior SD (for comparison only)” and move them to an appendix. But the main tables must be Bayesian summaries.

6) Numerical issues/table inconsistencies / implausible values were not resolved. The authors re-ran analyses and included a sensitivity table (Appendix Table 1) and several new result tables. They also discuss ESS and R-hat.

Large and inconsistent coefficients across methods remain. Example: the Metropolis table shows Ultrasound coefficient ≈ 4.89 (with CI 4.51–9.25) while HMC shows Ultrasound Y1 ≈ 0.28 (CI 0.08–0.48). For Y2, some full models report Ultrasound ≈ 12.51 in one table and ≈ 5.46 in another; intercepts sometimes are large and negative (−9.89) — these differences are large and unexplained.

The authors must:

State explicitly whether predictors were standardized/centered in each analysis. If not, do so and re-run or provide conversions so the coefficients are comparable.

Explain any parameterization differences between packages (e.g., different link or latent scale, log-scale reporting of variances, inverse-Wishart vs LKJ scaling).

7) Outdated dataset & clinical context. The authors added explicit mention of historical data provenance and an ethics statement indicating the original study had ethical approval and the current analyses used anonymized/deidentified records. They briefly discuss temporal limitations.

Strengthen the limitations paragraph by saying explicitly how changing clinical practice since 1988–89 might affect interpretation, and remove any wording implying clinical recommendations based solely on this dataset.

I pointed this out previously, and the authors failed to address it. Fix “Metropolis-Hastening” � Metropolis–Hastings throughout. There are still scattered typos and nonstandard terms (e.g., “random gradient” vs “random slope”). The authors stated they corrected many terms, but I still noticed a few. Please, carefully address the reviewer’s comments.

Next time, highlight revised texts with standard colors that allow reading. Bright green is not recommended.

.

Reviewer #1: **Yes:**Angel Alfonso García O'DianaAngel Alfonso García O'DianaAngel Alfonso García O'DianaAngel Alfonso García O'Diana

Reviewer #2: No

---

## [Author Response · Author response to Decision Letter 2]

5 Jan 2026

Dear Reviewer, we apologize for the oversight in our responses. Currently, we have tried to explain and address the necessary issues and to add more critical concepts raised by the editor/reviewers in the revised manuscript.

Note: The dataset used is repeated measurements from the same subjects had been taken on legs and leg sides; the label "1988/89" usually refers to the timing of the fieldwork, not the intent to measure change over two distinct years

Thank you.

---

## [Decision Letter · Decision Letter 2]

13 Feb 2026

Dear Dr. Ebrahim,

Thank you for submitting your manuscript to PLOS ONE. After careful consideration, we feel that it has merit but does not fully meet PLOS ONE’s publication criteria as it currently stands. Therefore, we invite you to submit a revised version of the manuscript that addresses the points raised during the review process.

We look forward to receiving your revised manuscript.

Kind regards,

Denekew Bitew Belay, Ph.D

Academic Editor

PLOS One

Journal Requirements:

Reviewers' comments:

Reviewer's Responses to Questions

**Comments to the Author**

Reviewer #2: All comments have been addressed

2. Is the manuscript technically sound, and do the data support the conclusions?

Reviewer #2: Yes

3. Has the statistical analysis been performed appropriately and rigorously?

Reviewer #2: Yes

4. Have the authors made all data underlying the findings in their manuscript fully available?

Reviewer #2: Yes

5. Is the manuscript presented in an intelligible fashion and written in standard English?

Reviewer #2: Yes

Reviewer #2: PONE-D-25-42936R2 Comment

The authors have made substantial and constructive revisions that have significantly improved the overall quality, clarity, and technical rigor of the manuscript. The responses to prior comments are generally appropriate and demonstrate careful consideration of the review process. Nevertheless, a small number of minor issues remain that should be addressed to ensure consistency, precision, and completeness before a final decision can be reached. Specific recommendations for these final minor adjustments are provided below.

1. Terminology and abbreviations must be introduced once at first occurrence and then used consistently without redefinition throughout the manuscript. This is critical in the manuscript, where repeated expansion of the same term disrupts readability and suggests insufficient editorial control. One example is the expression “Hamiltonian Monte Carlo (HMC)” that is constantly defined throughout the manuscript. For example, the paragraph after Table 3 starts and ends with “Hamiltonian Monte Carlo (HMC)”, followed by Table 4, which again reintroduces “Hamiltonian Monte Carlo (HMC)” as though it had not been defined earlier.

Consistent abbreviation usage will improve flow, reduce redundancy, and enhance the professional presentation of the manuscript. This should be revised throughout the manuscript for all other terms as well.

2. Captions could be improved. For example, the caption of Table 9 is not self-contained.

3. Please standardize the notation used for figure references throughout the manuscript. Both “Fig.” and “Fig” are currently used; only one format should be adopted consistently in accordance with the journal style guide (typically “Fig.” with a period). Consistent formatting contributes to a more professional and polished presentation.

4. Figure citation order in the main text must follow a strictly sequential progression. At present, the narrative flow is disrupted because the discussion moves from Fig. 4 directly to Fig. 9, while Figs. 5–8 are introduced later or not at all. Specifically, Fig. 9 appears in the second paragraph of Section 3.4 (“Evaluation of MCMC convergence diagnostics and conditional/marginal effects for the best-fitted model”), whereas Figs. 5–7 are first cited approximately four pages later (paragraph 3), and Fig. 8 does not appear to be referenced in the text. Figures should be introduced in numerical order (4 > 5 > 6 > 7 > 8 > 9) at the point where their content is first discussed. This reordering is essential to maintain logical continuity and to help readers follow the progression of results without confusion.

5. Revise the DOI to references and make sure they are all accessible to the readers of the paper. Some are below (I limit the below list to letters A-D only), just to show how critical this issue is.

i) Kallioinen, N., Paananen, T., Bürkner, P. C., & Vehtari, A. (2024). Detecting and diagnosing prior and likelihood sensitivity with power-scaling. Statistics and Computing, 34(1), 1– 27. https://doi.org/10.1007/S11222-023-10366-5/FIGURES/1

ii) Brooks, S., Gelman, A., Jones, G. L., & Meng, X. L. (2011). Handbook of Markov Chain Monte Carlo. Handbook of Markov Chain Monte Carlo, 1–592. https://doi.org/10.1201/B10905/HANDBOOK-MARKOV-CHAIN-MONTE-CARLO GALIN-JONES-XIAO-LI-MENG-ANDREW-GELMAN-STEVE-BROOKS/RIGHTS AND-PERMISSIONS

iii) Chien, Y. F., Zhou, H., Hanson, T., & Lystig, T. (2023). Informative g-Priors for Mixed Models. Stats, 6(1), 169–191. https://doi.org/10.3390/STATS6010011/S1 (remove “/S1”)

iv) Cook, S. R., Gelman, A., & Rubin, D. B. (2006). Validation of software for Bayesian models using posterior quantiles. Journal of Computational and Graphical Statistics, 15(3), 675–692. https://doi.org/10.1198/106186006X136976;REQUESTEDJOURNAL:JOURNAL:UCG S20;WGROUP:STRING:PUBLICATION (correct = https://doi.org/10.1198/106186006X136976 )

v) Depaoli, S., Winter, S. D., & Visser, M. (2020). The Importance of Prior Sensitivity Analysis in Bayesian Statistics: Demonstrations Using an Interactive Shiny App. Frontiers in Psychology, 11, 608045. https://doi.org/10.3389/FPSYG.2020.608045/BIBTEX (correct is https://doi.org/10.3389/fpsyg.2020.608045 )

.

Reviewer #2: **Yes:**Cebastien GuembouCebastien GuembouCebastien GuembouCebastien Guembou

---

## [Author Response · Author response to Decision Letter 3]

22 Feb 2026

Dear Reviewer #2, thank you for your critical observation and constructive comments on our work. We tried to edit all the necessary comments and suggestions. The reviewer responses and revised manuscript are attached.

---

## [Decision Letter · Decision Letter 3]

3 Mar 2026

Dear Dr. Ebrahim,

Thank you for submitting your manuscript to PLOS ONE. After careful consideration, we feel that it has merit but does not fully meet PLOS ONE’s publication criteria as it currently stands. Therefore, we invite you to submit a revised version of the manuscript that addresses the points raised during the review process.

We look forward to receiving your revised manuscript.

Kind regards,

Denekew Bitew Belay, Ph.D

Academic Editor

PLOS One

Journal Requirements:

Reviewers' comments:

Reviewer's Responses to Questions

**Comments to the Author**

Reviewer #2: (No Response)

2. Is the manuscript technically sound, and do the data support the conclusions?

Reviewer #2: Yes

3. Has the statistical analysis been performed appropriately and rigorously?

Reviewer #2: Yes

4. Have the authors made all data underlying the findings in their manuscript fully available?

Reviewer #2: Yes

5. Is the manuscript presented in an intelligible fashion and written in standard English?

Reviewer #2: No

Reviewer #2: I appreciate the authors’ effort in revising the manuscript and their willingness to address the previous comments. The work has improved, and I would like to see it progress further toward acceptance. However, several minor issues remain and should be carefully addressed to ensure the manuscript meets the expected standards of scientific presentation and editorial quality.

First, the citation format within the text requires systematic correction. There should be a space between the preceding word and the citation number. For example, “structures[7].” should be revised to “structures [7].” This formatting issue appears throughout the manuscript and should be corrected consistently.

Second, the narrative integration of references is often inappropriate. Expressions such as “[11] used R” or “. [19] demonstrated the usefulness” are not acceptable in standard scientific writing. Instead, references should be incorporated into the sentence in a grammatically correct manner by naming the authors, followed by the citation number. For example, this should be written as “Lemoine (2019) used R … [11].” or “Tate and Pituch (2007) demonstrated the usefulness … [19].” This improves readability, attribution clarity, and professionalism. The authors should revise the manuscript throughout to ensure all references are introduced properly.

Third, and most concerning, the DOI and reference corrections requested previously have not been adequately implemented. Although the authors indicated that these had been revised, numerous references in the revised manuscript still contain incorrect, duplicated, or non-functional DOI links. This suggests insufficient attention during revision. Correct and accessible DOIs are essential to allow readers to verify and consult the cited literature. The following critical issues must be corrected:

• Reference [12]: Incorrect DOI. The correct DOI is: https://doi.org/10.1515/jqas-2018-0106

• Reference [13]: Incorrect DOI. The correct DOI is: https://doi.org/10.1186/s12874-024-02333-z

• References [30] and [31]: Incorrect DOI, same issue as [12]

• References [40] and [55]: Incorrect DOI, same issue as [12]

• References [65], [66], and [78]: Incorrect DOI, same issue as [12]

• Reference [41]: Duplicate DOI

• Reference [53]: Same duplicated DOI as [41]

• Reference [60]: DOI and reference details require verification and correction

The authors should carefully review the entire reference list, verify each DOI individually, ensure that each DOI points to the correct article, remove duplicate entries, and confirm that all references are accurate and accessible.

Previous comment #1: stating "Terminology and abbreviations must be introduced once at first occurrence and then used consistently without redefinition throughout the manuscript. This should be revised throughout the manuscript for all other terms as well."

The manuscript does not follow standard conventions regarding the introduction and consistent use of abbreviations. Terminology and abbreviations should be defined at their first occurrence and used consistently thereafter without redefinition. This issue persists despite my previous request for correction. In particular, the abbreviation MCMC is used in the abstract without definition, appears again in the first paragraph of the Introduction, and is only defined later in the fourth paragraph of that section. This delayed definition disrupts readability and forces the reader to proceed without knowing the meaning of a key methodological term. The current structure is therefore awkward and should be revised so that MCMC is defined at its first occurrence in the abstract and then used consistently throughout the manuscript. Similar attention should be applied to all other abbreviations to ensure clarity and proper scientific presentation.

Overall, these issues are correctable but require careful and thorough revision. Addressing them properly will significantly improve the clarity, professionalism, and reliability of the manuscript and facilitate its progression toward acceptance.

.

Reviewer #2: **Yes:**Cebastien GuembouCebastien GuembouCebastien GuembouCebastien Guembou

---

## [Author Response · Author response to Decision Letter 4]

11 Mar 2026

Dear reviewer #2 and Editor,

Thank you for your critical review/ edition of our manuscript. We reviewed and edited all the required suggestions and the updated manuscript and one-by one point responses are attched.

---

## [Editor Report · Decision Letter 4]

18 Mar 2026

Bayesian Hierarchical Models for Multivariate Mixed Responses with Repeated Measures: A Case Study in Arterial Occlusive Disease

PONE-D-25-42936R4

Dear Dr. Ebrahim,

We’re pleased to inform you that your manuscript has been judged scientifically suitable for publication and will be formally accepted for publication once it meets all outstanding technical requirements.

Kind regards,

Denekew Bitew Belay, Ph.D

Academic Editor

PLOS One
---

## [Editor Report · Acceptance letter]

PONE-D-25-42936R4

PLOS One

Dear Dr. Ebrahim,

I'm pleased to inform you that your manuscript has been deemed suitable for publication in PLOS One. Congratulations! Your manuscript is now being handed over to our production team.

Kind regards,

on behalf of

Dr. Denekew Bitew Belay

Academic Editor

PLOS One